# On the Theory of Continual Learning with Gradient Descent for Neural Networks

**Hossein Taheri** [1]  **Avishek Ghosh** [2]  **Arya Mazumdar** [3]

## Abstract

Continual learning, the ability of a model to adapt to an ongoing sequence of tasks without forgetting earlier ones, is a central goal of artificial intelligence. To better understand its underlying mechanisms, we study the limitations of continual learning in a tractable yet representative setting. Specifically, we analyze one-hidden-layer quadratic neural networks trained by gradient descent on a sequence of XOR-cluster datasets with Gaussian noise, where different tasks correspond to clusters with orthogonal means. Our analysis is based on a tight characterization of gradient descent dynamics for the training loss, which yields explicit bounds on the rate of train-time forgetting as functions of the number of iterations, sample size, number of tasks, and hidden-layer width. We then leverage an algorithmic stability framework to bound the generalization gap, leading to corresponding guarantees on test-time forgetting. Together, our results provide the first closed-form guarantees for forgetting in continual learning with neural networks and show how key problem parameters jointly govern forgetting dynamics. Numerical experiments corroborate our theoretical results.

## 1. Introduction

### 1.1. Motivation

Gradient-based methods are the dominant paradigm for training neural networks, and recent advances in learning theory have shown that such models can efficiently learn a wide range of data distributions through empirical risk minimization (ERM). However, in many real-world applications, data are not presented all at once but arrive sequentially in a non-stationary fashion, requiring the learner to maintain performance on past tasks while acquiring new capabilities. In such cases, a learning model must be continually learnable, meaning it should retain previously acquired knowledge when trained on new tasks. On the other hand, various learning systems, including deep learning architectures, can be prone to *catastrophic forgetting*, that is, updating a model on new data causes a dramatic drop in performance on previously learned tasks (McCloskey & Cohen, 1989; Goodfellow et al., 2013). The goal of continual (lifelong) learning is to experience minimal forgetting when incorporating new information, even without retraining on old data.

Despite the importance of continual learning for modern AI systems, the theoretical works studying the mechanisms behind forgetting are limited. A growing body of recent works have developed guarantees for continual learning in linear models and generalized linear models, often under realizability conditions. In these settings, a single linear predictor can interpolate data from all tasks, and forgetting can be controlled either through implicit bias or exact closed-form ERM solutions (Evron et al., 2023; 2022; Banayeeanzade et al., 2024; Li et al., 2025b). While these results provide valuable insight into the mechanisms of forgetting, they rely on linear structure crucially - in particular on the fact that statistical and optimization properties of linear models are already very well-understood.

Our work goes beyond linear models by providing the first closed-form forgetting guarantees for neural networks trained by gradient descent. Our analysis reveals that for neural networks, forgetting is governed by the interplay between sample size of subsequent tasks, hidden-layer width, and early stopping, and that none of these factors alone is sufficient to eliminate forgetting, which does not have an analog in previous works. Our results explicitly track the evolution of the weights across tasks and show that, despite tasks being independent and orthogonal, *learning later tasks can suppress forgetting* depending on how their sample sizes and training horizons scale. This is verified by our empirical observations in several setups that increasing the

---

[1]Department of Computer Science and Engineering, University of California, San Diego, USA. [2]Department of Computer Science and Engineering, Indian Institute of Technology, Bombay, India. [3]Halicioğlu Data Science Institute, University of California, San Diego, USA. Correspondence to: Hossein Taheri <htaheri@ucsd.edu>, Avishek Ghosh <avishek_ghosh@iitb.ac.in>, Arya Mazumdar <arya@ucsd.edu>.

*Proceedings of the $43^{rd}$ International Conference on Machine Learning*, Seoul, South Korea. PMLR 306, 2026. Copyright 2026 by the author(s).

dataset size of later tasks can significantly reduce or vanish forgetting of earlier ones. Moreover, unlike prior works, our framework yields guarantees for both *train-time* and *test-time* forgetting. We decompose test-time forgetting into a training-loss component and a delayed generalization gap induced by intermediate tasks. By combining a fine-grained analysis of gradient descent dynamics with an algorithmic stability argument tailored to continual learning, we derive explicit conditions under which both terms vanish. This decomposition reveals that forgetting may persist unless the network width, number of samples, and training horizon scale appropriately with the key problem parameters such as the number of tasks.

While several recent works study the convergence and sample-complexity of gradient methods on neural networks under stylized data distributions, they are largely restricted to single-task settings, leaving the role of optimization dynamics in continual learning poorly understood (for some examples see (Du et al., 2019; Bartlett et al., 2021; Damian et al., 2022; Abbe et al., 2022)). In this work, we present several results on the performance of gradient descent in neural networks in the kernel regime for scenarios where there is a stream of tasks on which the model is sequentially trained. Focusing on unregularized empirical risk minimization, we identify regimes in which gradient descent, without explicit continual-learning regularization, achieves arbitrarily small train-time and test-time forgetting. To this end, *we study a representative data model based on XOR clusters, a canonical and non-linearly separable model that underlies multi-index and parity learning problems, and characterize the sample, iteration, and model-complexity requirements for successful continual learning*. Although our focus is on the XOR cluster distribution, our approach is applicable to various data distributions, and we expect similar insights can be derived for various single- or multi-index models. We further show that in the kernel regime, the regularized continual learning algorithm does not mitigate forgetting, as it is equivalent to unregularized ERM with a rescaled step size. Consequently, our theoretical predictions continue to hold in the presence of regularization.

To establish these results, we develop a principled analysis that integrates optimization dynamics with generalization arguments, allowing us to explicitly characterize how training, data, and model parameters jointly govern forgetting in continual learning.

**Techniques and Contributions.** Our method is based on the decomposition of the test-time forgetting error into two terms based on forgetting in training loss and the delayed generalization gap caused by intermediate learning tasks. First, we bound the generalization gap by an argument based on algorithmic stability (Bousquet & Elisseeff, 2002; Lei & Ying, 2020; Richards & Rabbat, 2021; Taheri et al., 2025)

tailored to our set-up of continual learning with neural nets, which leads to conditions on the network width and the number of iterations and samples to achieve a small generalization error after learning independent intermediate tasks from distinct distributions. Our results reveal that generalization gap for continual learning with neural networks is impacted by the training loss of later tasks (as in Thm B.1) or number of tasks (as in Thm 2.3) which is new compared to single-task analyses. We then use a data-specific argument to formulate the evolution of the learned weights throughout the gradient descent steps to bound the training loss and forgetting in Theorems 2.1-2.2. In particular, we first consider an asymptotic regime where for both the sample size and network size $m, n \rightarrow \infty$. The critical observation here is that in this regime, for every task, the gradient at initialization is in the correct direction, and with sufficient number of GD steps, the train loss and the amount of forgetting(i.e., the increase in training loss caused by learning later tasks) are asymptotically zero. As a result of this and with concentration bounds for finite $n$ and $m$, we are able to characterize the rate of forgetting based on these parameters. This differs from the existing analyses of neural nets for single-task classification setups in the lazy regime which are mainly based on class margin (Nitanda et al., 2019; Ji & Telgarsky, 2020; Taheri & Thrampoulidis, 2024). To the best of our knowledge, our results are the first closed-form guarantees for the train and test performance of continual learning methods when using neural networks and they predict several of the empirical observations on the role of training-set size and over-parameterization. In summary, our contributions are the following:

- We study the role of key problem parameters such as sample complexity and over-parameterization for continual learning with neural networks. Specifically, for the $d$-dimensional XOR-cluster dataset, we derive explicit bounds on train-time forgetting after learning $K$ subsequent tasks, showing it scales as $\widetilde{O}(\eta T \frac{\sqrt{K}}{d\sqrt{n}} + \eta T \frac{\sqrt{K}}{d^2 \mathrm{polylog}(d)} + \eta^2 T^2 \frac{K^2}{\sqrt{m}})$, where $T$ and $n$ denote the number of gradient descent iterations and samples per task, respectively, and $m$ is the hidden-layer width.

- We characterize the sample and computational requirements for successful continual learning. In particular, we show that choosing $n = \widetilde{\Theta}(d^2 K), m = \Theta(d^8 K^4), T = \widetilde{\Theta}(d^2)$ is sufficient to ensure uniformly small training loss across all $K$ tasks, yielding vanishing train-time forgetting.

- By decomposing test-time forgetting into a train-time forgetting term and a delayed generalization gap, and bounding the latter via an algorithmic-stability analysis tailored to continual learning, we show that the above scaling of $n, m$, and $T$ also leads to vanishing test-time

forgetting.

- We validate our theoretical predictions empirically. Numerical experiments on different losses, activation functions, datasets, and architectures corroborate our analysis and demonstrate that the identified roles of sample size, over-parameterization, and early stopping persist in different settings.

Overall, our results identify a positive regime for continual learning with neural nets and provide first known explicit conditions for successful continual learning with plain gradient methods, going beyond the linear models considered in the literature.

## 1.2. Related Works

Despite extensive empirical and algorithmic progress in continual learning, a principled theoretical understanding of catastrophic forgetting in neural networks trained by gradient descent is still largely missing. The main algorithms for continual learning are based on functional (or architectural) regularization (Li & Hoiem, 2017; Kirkpatrick et al., 2017; Sharif Razavian et al., 2014) or experience replaying (Schaul et al., 2016; Rolnick et al., 2019). In order to mitigate forgetting, regularization-based methods enforce the new solutions to remain close to solutions to previous tasks. On the other hand, it has been hypothesized that the network width has a similar impact (Graldi et al., 2024), since increasing the width enables the network to operate in the lazy/kernel regime where it is known that the network's weights do not travel a significant distance from their initialization point and the features remain constant during training (Jacot et al., 2018; Ghorbani et al., 2019). It is therefore natural to ask to what extent the width helps continual learning. Few works have dealt with this question. In particular, the impact of the width of the network on continual learning was studied in (Guha & Lakshman, 2024; Graldi et al., 2024; Mirzadeh et al., 2022a;b; Wenger et al., 2023). For instance, (Mirzadeh et al., 2022a;b) empirically observed the impact of width in improving catastrophic forgetting and noticed that increasing the width always mitigates forgetting. However, (Wenger et al., 2023) claimed that such improvements vanish when the network is trained for a sufficiently large number of iterations until convergence. More recently, (Graldi et al., 2024) attempted to resolve the issue claiming that improvements only happen in the kernel regime, where there is early stopping to avoid weights moving a significant distance from their initialization. Our theoretical and empirical results on the impact width also verify the benefits of width in the kernel regime with early stopping.

(Guha & Lakshman, 2024) showed analytically through a general argument that increasing the width helps continual learning, although the improvements shrink as width grows. The dependence on width in their bound is not explicitly determined, and moreover, the bound does not depend on the underlying algorithm or number of samples. In contrast, our analysis is algorithm-dependent and yields closed-form bounds, explicitly highlighting the roles of different problem parameters such as over-parameterization in test-time forgetting.

Perhaps the closest works to ours are (Doan et al., 2021; Bennani et al., 2020; Lee et al., 2021); see also (Karakida & Akaho, 2021), which derived general expressions to characterize forgetting in neural networks in the lazy regime. A recent work by (Li et al., 2025a) focuses specifically on CNNs in a multi-view data model and characterizes forgetting. (Benjamin et al., 2024) approach uses an "ensemble/NTK" perspective treating networks in the lazy regime and gives a reinterpretation of continual learning. (Cao et al., 2022) derived sample complexity of continually learning linear models and GLMs. (Andle & Yasaei Sekeh, 2022) focus on layer-wise information flow and develop a probabilistic theory for CL performance across layers. However, the prior works discussed above do *not* lead to closed-form bounds and are applicable for different models such as CNNs, while our results yield the first bounds for a multi-index model learned by neural nets.

Another related line of work has focused on linear classification/regression in the realizable regime, where a single linear solution can interpolate data from all tasks (Goldfarb & Hand, 2023; Lin et al., 2023; Evron et al., 2023; Banayeeanzade et al., 2024). In particular, (Evron et al., 2023) analyzed catastrophic forgetting through the lens of implicit bias in linear classification across various setups, including cyclic and random task orderings. Since these approaches are distribution-independent, they do not reveal the role of sample complexity or over-parameterization. In contrast, we adopt a more practical perspective by examining sample complexity, early stopping, and the effects of over-parameterization in a stylized neural network setting.

## Notation

We use the standard complexity notation $\lesssim, o(\cdot), O(\cdot), \Theta(\cdot), \Omega(\cdot)$ and denote $\widetilde{o}(\cdot), \widetilde{O}(\cdot), \widetilde{\Theta}(\cdot), \widetilde{\Omega}(\cdot)$ to hide poly-logarithmic factors in $d$. The subscripts in $O_d(\cdot), o_d(\cdot)$ denote the dependence on the parameter $d$. We use $\| \cdot \|$ for the $\ell_2$ norm of vectors. We denote $[n] := \{1, 2, \cdots, n\}$. The expectation and probability with respect to the randomness in $\mathcal{D}$ are denoted by $\mathbb{E}_{\mathcal{D}}[\cdot], \Pr_{\mathcal{D}}(\cdot)$. The gradient of the model $\Phi : \mathbb{R}^{p \times d} \to \mathbb{R}$ with respect to the first input (weights) is denoted by $\nabla \Phi$.

# 2. Main Results

## 2.1. Problem Setup

### 2.1.1. GRADIENT-BASED CONTINUAL LEARNING WITH NEURAL NETWORKS

We consider the problem of sequentially learning $K$ independent tasks, where each task is trained in isolation. Specifically, for the $k$-th task, we perform $T$ iterations of gradient descent using a dataset of $n$ training samples. The objective of task $k$ is defined as

$$\widehat{F}(w, \mathcal{D}_k) = \frac{1}{n} \sum_{i=1}^{n} f\big(y_i\, \Phi(w, x_i)\big),$$

where $\mathcal{D}_k = \{(x_i, y_i)\}_{i=1}^{n}$ denotes the set of training examples for task $k$, and the mapping $\Phi$ represents a two-layer neural network with $m$ hidden neurons and activation $\phi$, given by $\Phi(w, x) = \frac{1}{\sqrt{m}} \sum_{i=1}^{m} a_i\, \phi(x^\top w_i)$. Throughout the paper, we assume that the output layer coefficients $a_i \in \{\pm 1\}$ are fixed, let $f$ be the hinge-loss and we focus on the case of quadratic activation where $\phi(t) = t^2/2$. For convenience, we denote the empirical loss for task $k$ by $\widehat{F}_k(w) := \widehat{F}(w, \mathcal{D}_k)$, and the corresponding population (test) loss by $F_k(w) := F(w, \mathcal{P}_k) = \mathbb{E}_{(x,y) \sim \mathcal{P}_k}\big[f\big(y\, \Phi(w, x)\big)\big]$, where the expectation is taken over the test-set distribution $\mathcal{P}_k$.

The complete continual learning procedure is summarized in Algorithm 1. We initialize the parameter vector $w_1^{(0)}$ from a standard Gaussian distribution, $w_1^{(0)} \sim \mathcal{N}(0, I_p)$, where $p = md$ is the total number of trainable parameters in the first layer. For each task $k \in \{1, \dots, K\}$, we train the network starting from initialization $w_{k-1}^{(T)}$ for $T$ gradient descent updates on $\widehat{F}_k$. The resulting vector after finishing the training on task $k$ is denoted by $w_k := w_k^{(T)} := w_{k+1}^{(0)}$, and it serves as the initialization for the subsequent task $k+1$. After processing all $K$ tasks, the algorithm outputs the final parameter vector $w_K$, which contains the accumulated knowledge obtained from the entire sequence of tasks.

### 2.1.2. XOR CLUSTER DATASET

Consider data according to the XOR cluster distribution with Gaussian noise where $x \in \mathbb{R}^d, y \in \{\pm 1\}$ and

$$x \sim \begin{cases} \frac{1}{2}\mathcal{N}(\mu_+, \sigma^2 I_d) + \frac{1}{2}\mathcal{N}(-\mu_+, \sigma^2 I_d) & \text{if } y = 1, \\ \frac{1}{2}\mathcal{N}(\mu_-, \sigma^2 I_d) + \frac{1}{2}\mathcal{N}(-\mu_-, \sigma^2 I_d) & \text{if } y = -1, \end{cases} \tag{1}$$

where $\mu_+ \perp \mu_-$, and $\Pr[y = 1] = \Pr[y = -1] = 1/2$. The XOR cluster and its Boolean variant (known as parities) have been extensively studied in the deep learning theory literature (Wei et al., 2019; Refinetti et al., 2021; Xu et al., 2024; Telgarsky, 2023; Taheri & Thrampoulidis, 2024; Glasgow, 2024). In particular, the XOR model is a representative

---

**Algorithm 1** Continual Learning with Gradient Descent

1: **Input:** Number of tasks $K$, number of steps per task $T$, learning rate $\eta$
2: **Output:** Final model parameters $w_K$
3: Initialize model parameters $w_1^{(0)} \sim \mathcal{N}(0, I_p)$
4: **for** $k = 1$ **to** $K$ **do**
5:      Load task-specific dataset $\mathcal{D}_k$
6:      **for** $t = 0$ **to** $T - 1$ **do**
7:          Sample a (mini or full) batch $\mathcal{B}_t \subseteq \mathcal{D}_k$
8:          $w_k^{(t+1)} \leftarrow w_k^{(t)} - \eta \nabla \widehat{F}(w_k^{(t)}; \mathcal{B}_t)$
9:      **end for**
10:      Set $w_{k+1}^{(0)} \leftarrow w_k := w_k^{(T)}$
11: **end for**
12: **return** $w_K := w_K^{(T)}$

---

instance of multi-index models, which have recently been used to investigate the sample complexity of neural network learning (Damian et al., 2022; Ba et al., 2022; Abbe et al., 2022). For this distribution, we show that $d^2$ samples and $d^4$ neurons are sufficient to achieve near zero train and test loss (see Prop. A.1 in Appendix A).

For the continual learning setup we consider a stream of $K$ tasks, where each task is generated according to the XOR cluster dataset, that is, for task $k$:

$$x \sim \begin{cases} \frac{1}{2}\mathcal{N}(\mu_+^k, \sigma^2 I_d) + \frac{1}{2}\mathcal{N}(-\mu_+^k, \sigma^2 I_d) & \text{if } y = 1, \\ \frac{1}{2}\mathcal{N}(\mu_-^k, \sigma^2 I_d) + \frac{1}{2}\mathcal{N}(-\mu_-^k, \sigma^2 I_d) & \text{if } y = -1. \end{cases} \tag{2}$$

This dataset serves as a representative example of a realizable problem that is well-suited for analyzing neural networks, specially for continual learning where different tasks correspond to different clusters of Gaussian data. We assume that $\mu_+^k$ and $\mu_-^k$ are mutually orthogonal for all $k \in [K]$, with $\|\mu_+^k\| = \|\mu_-^k\| = \Theta(\frac{1}{\sqrt{d}})$, balanced labels $\Pr[y = 1] = \Pr[y = -1] = 1/2$, and noise level $\sigma = \Theta(\frac{1}{\log^c(d)\sqrt{d}})$ for some universal positive constant $c$. The orthogonality assumption reflects the fact that tasks are uncorrelated. We note that our analysis can be extended to the more general case where the mean vectors are not orthogonal between tasks, by introducing cross-task interactions that significantly complicate the bound. We further assume that the number of tasks grows at most poly-logarithmically with the data dimension, i.e., $K = \widetilde{O}_d(1)$.

### 2.1.3. A DECOMPOSITION OF FORGETTING ERROR

Let $w_k$ denote the weights after training with data from task $k$ for some $k \in [K]$. *Test-time forgetting* is measured by the increase in test loss for the $k$th task after training on $K - k$ subsequent tasks:

Test-time Forgetting: $\mathcal{F}_{k,K}^{\text{ts}} := F_k(w_K) - F_k(w_k)$.

We can decompose the test-time forgetting as follows:

$$\mathcal{F}_{k,K}^{\text{ts}} = [F_k(w_K) - \widehat{F}_k(w_K)] + [\widehat{F}_k(w_K) - \widehat{F}_k(w_k)]$$
$$+ [\widehat{F}_k(w_k) - F_k(w_k)].$$

In the interpolating regime where the network can achieve zero training loss, we can drop the last term and bound the test-time forgetting based on *delayed generalization gap* and *training loss*:

$$\mathcal{F}_{k,K}^{\text{ts}} \leq \underbrace{\left[\widehat{F}_k(w_K) - \widehat{F}_k(w_k)\right]}_{\text{Train-time forgetting } \mathcal{F}_{k,K}^{\text{tr}}} + \underbrace{\left[F_k(w_K) - \widehat{F}_k(w_K)\right]}_{\text{Delayed generalization gap } \mathcal{F}_{k,K}^{\text{gen}}}$$
$$(3)$$

In the following section, we discuss each term separately. When combined, these will give an upper bound on the expected test-time forgetting.

### 2.2. Train and Test-time Forgetting Bounds

The following theorem provides closed-form bounds on the train-time forgetting of task $k$ after learning the subsequent $K - k$ tasks (for a total of $K$ tasks). We assume the hinge loss, $f(u) = \max\{1 - u, 0\}$, and adopt the data distribution specified in Eq. 2. The proofs for the theorems in this section are deferred to the appendix.

**Theorem 2.1** (Train-time forgetting). *Consider the $d$-dimensional XOR cluster dataset with $K$ tasks and assume gradient descent with $\eta T = \Theta(d^2)$ iterations and $n = \widetilde{\Theta}(d^2 K)$ samples for each subsequent task trained by a neural net with $m = \widetilde{\Omega}(d^8 K^4)$ hidden neurons. Then, with high probability, the train-time forgetting is $\mathcal{F}_{k,K}^{\text{tr}} = o_d(1)$. In particular, with probability $1 - \delta$, we have:*

$$|\mathcal{F}_{k,K}^{\text{tr}}| := |\widehat{F}_k(w_K) - \widehat{F}_k(w_k)| \qquad (4)$$
$$= \widetilde{O}\left(\eta T \frac{\sqrt{K-k}}{d\sqrt{n}} + \eta T \frac{\sqrt{K-k}}{d^2 \operatorname{poly}\log(d)} + \eta^2 T^2 \frac{K^2}{\sqrt{m}}\right),$$

*where $\widetilde{O}(\cdot)$ hides logarithmic factors in $n, T$ and $1/\delta$.*

The first and third terms in Eq. 4 capture the effects of sample size and hidden-layer width. Importantly, neither factor alone is sufficient to eliminate train-time forgetting. However, with sufficiently large $n$ and $m$, as stated in the theorem, we obtain a forgetting rate $\mathcal{F}_{k,K}^{\text{tr}} = O(1/\operatorname{poly}\log(d)) = o_d(1)$. Here, $n$ denotes the sample size of datasets learned after task $k$. Although these subsequent tasks are independent of and orthogonal to task $k$ (tasks are IID with orthogonal means), their larger training sets nevertheless reduce noise due to sampling and enhance the overall continual learning process. Our experiments in Section 3, conducted across different activation functions, loss functions, and datasets under various problem settings,

empirically confirm the theoretical roles of network width, sample size, and the number of tasks.

We note that the early-stopping choice $\eta T = \widetilde{\Theta}(n)$ is standard in the deep learning literature, particularly in the interpolation regime for single-task settings (Ji & Telgarsky, 2020; Lei & Ying, 2020). As the following theorem demonstrates, under this choice the training loss remains uniformly small across all tasks.

**Theorem 2.2** (Train error in continual learning). *Let the assumptions of Theorem 2.1 hold. Then, after $KT$ iterations of GD, with high probability, the misclassification train error and train loss are $o_d(1)$ uniformly for all $K$ tasks.*

The proofs of Theorems 2.1-2.2 are deferred to App. C. A combination of these theorems yields sufficient conditions for successful continual learning as measured by training performance. We remark that the proof of both theorems (up to calculations related to the model-output's equations in the appendix in Eq.14 or forgetting equation in Eq.16) holds for a broad family of data distributions. The parts of the analysis that specialize to the XOR-cluster distribution primarily arise when deriving explicit closed-form expressions for the model output and for characterizing closed-form bounds for forgetting. Therefore, although our main theorems so far are stated for orthogonal xor-cluster tasks, the proof strategy is not tied exclusively to this setting. As mentioned in Section 2.1.2, this choice of data is motivated by the literature on single/multi-index models in deep learning theory. Specializing to this class of data also makes it possible to isolate the role of key problem parameters, such as width, sample size, number of tasks, and training horizon, in a transparent way. More generally, the train-time forgetting analysis suggests that for general data (cf. Eqs. 16-17 in the appendix):

$$\left|\widehat{F}_k(w_K) - \widehat{F}_k(w_k)\right| =$$
$$\left|\frac{1}{n} \sum_{x_k} \eta T x_k^\top \left(\sum_{j=k+1}^K A_j\right) x_k\right| + O\left(\frac{\|w_K - w_0\|^2}{\sqrt{m}}\right),$$

where $A_j := \frac{1}{n} \sum_{v=1}^n y_j^v x_j^v (x_j^v)^\top$, and $\{(x_j^v, y_j^v)\}_{v \in [n]}$ denotes the training data for task $j$. Theorem 2.1 is obtained by specializing this characterization to the XOR-cluster model with Gaussian noise, where orthogonality removes cross-task interference terms and yields a clean closed-form dependence on the problem parameters. For more general data distributions, additional correlation terms appear, and the resulting expressions become more cumbersome; nevertheless, we expect the same framework to provide similar qualitative insights into how width, sample size, data noise and task overlap affect forgetting.

Our next result derives the delayed generalization gap (as

defined in Eq.3) for almost any data distribution. In fact, it also shows that the derived scalings for $m, n$ and $T$ in Thm. 2.1 are also sufficient for good continual *test-time* performance for the XOR cluster dataset.

**Theorem 2.3** (Delayed generalization gap). *Assume the loss function is 1-Lipschitz and 1-smooth. Then, the expected delayed generalization gap satisfies,*

$$\mathcal{F}^{\mathrm{gen}}_{k,K} := \mathbb{E}_{\mathcal{D}_k}\left[ F_k(w_K) - \widehat{F}_k(w_K) \right] \lesssim \frac{\eta T\, e^{\frac{\eta T(K-k+1)}{\sqrt{m}}}}{n},$$

*where the expectation above is taken with respect to the randomness in the training examples of task $k$.*

*Remark* 2.4 (Test-time forgetting). The bound above holds on expectation over the choice of training set for task $k$ and it holds with probability 1 with respect to the randomness in the training set of subsequent tasks. Note that the bound decays with the rate $1/n$ and given sufficiently large width, it is linearly proportional to the number of iterations. While the bound holds for almost any data distribution, for the XOR-cluster distribution with the training loss guarantees from Thms. 2.1-2.2, we find that with $n = \widetilde{\Theta}(d^2 K) = \widetilde{\Theta}(\eta T)$ samples and with $m = \widetilde{\Omega}(d^8 K^4)$, it holds $\mathcal{F}^{\mathrm{gen}}_{k,K} = o_d(1)$ resulting in vanishing test-time forgetting in view of Eq. 3. Finally, we note – as the proof shows – training occurs within the linear region of the hinge loss, which allows us to combine the results of the previous theorems despite the smoothness assumption on the loss in Theorem 2.3.

*Remark* 2.5. The results of Thms 2.1-2.3 provide the first closed-form characterization of forgetting for neural networks trained sequentially in the kernel regime and identify an explicit regime in which continual learning is possible with plain sequential gradient descent. Particularly, the results so far show that unregularized sequential gradient descent can simultaneously achieve arbitrarily small forgetting and small test error across all tasks under a regime where the data noise is sufficiently small (i.e., $\sigma = O(\frac{1}{\mathrm{poly} \log(d)\sqrt{d}})$) and the tasks are separated (orthogonal clusters' means), providing sufficient conditions for successful continual learning with GD even without regularization or data-replay.

*Remark* 2.6 (Improved rates). The proof of Theorem 2.3 is deferred to Appendix B. In particular, we first prove a specific version of this theorem (stated as Theorem B.1 in the appendix) under additional conditions on the loss. We note that while Theorem 2.3 is sufficient to ensure that test-time forgetting is asymptotically small under the parameter scalings considered above, its dependence on the training horizon $T$ can become a bottleneck in more general settings. In particular, one typically expects the generalization gap to depend only weakly on the number of gradient-descent iterations. To address this limitation, we introduce additional assumptions on the loss function (satisfied, in particular, by the logistic loss) that yield a sharper control of the stability of the gradient-descent trajectory. Under these conditions,

we establish an improved generalization bound in Theorem B.1 in the appendix, where the dependence on $T$ is reduced to *polylogarithmic*. This refinement leads to improved conditions on the network width and yields a tighter bound on the delayed generalization gap based on $T$. We also note that, the qualitative trends predicted by the theory already appear at substantially smaller widths and different training horizons in our experiments in Section 3, suggesting that our results are valid beyond the derived scalings in the theorems.

## 2.3. Regularized Continual Learning

It is natural to ask whether regularization can improve the scalings derived in the last section. We consider the regularized continual learning algorithm (e.g., (Aljundi et al., 2017; Kirkpatrick et al., 2017; Lewkowycz & Gur-Ari, 2020)) with parameter $\lambda$ where for each task $k \geq 2$, the objective is to minimize the following,

$$\min_{w}\ \widehat{F}_k(w) + \frac{\lambda}{2}\|w - w_{k-1}\|^2. \tag{5}$$

The regularization parameter $\lambda$ can be chosen to be fixed, time-varying or data-dependent (Evron et al., 2023; Lewkowycz & Gur-Ari, 2020; Kirkpatrick et al., 2017). In the next proposition, we consider the fixed $\lambda$ in order to study the effects of regularization on the GD iterates. Our next result shows that in the linearized regime (i.e., the infinite-width regime) where the network output can be written as a first-order approximation around initialization, the regularized continual learning problem is effectively equivalent to unregularized minimization with a time-varying step-size.

**Proposition 2.7** (Regularized continual learning). *Consider the regularized continual learning problem Eq.5 in the linearized regime, with the same setup as Theorem 2.1. The iterates of this algorithm with step-size $\eta$ are equivalent to unregularized continual learning with step-size $\widetilde{\eta}_T$ for any task $k \geq 2$, where we define $\widetilde{\eta}_T := \frac{\alpha_T \eta}{T}$ and $\alpha_T := \frac{1-(1-\eta\lambda)^T}{\eta\lambda}$.*

Hence, as $T$ increases, the effective step-size decreases, preventing iterations from moving a significant distance from the solution of previous task. In particular, Prop. 2.7 shows that the considered $L_2$-type regularization around the previous-task solution is equivalent to unregularized continual learning with a different effective step size.

Consequently, within the kernel regime studied in the last section, such regularization does not fundamentally improve the forgetting and generalization scalings derived in our analysis. At the same time, this does not preclude the possibility that other mechanisms (such as architectural modifications, or regularization under regimes with stronger feature learning) may significantly improve continual learning performance.

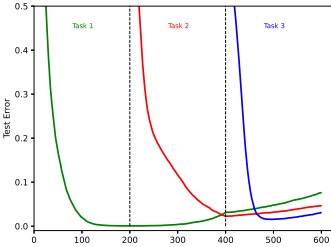 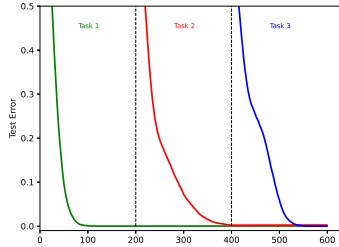

*Figure 1.* Classification test error for each task vs. iterations for the XOR cluster with 3 tasks trained on a quadratic network with $n = 2500$ (left) and $n = 5000$ (right) training samples per task. Increasing sample-size can lead to *vanishing forgetting and test error* on all tasks.

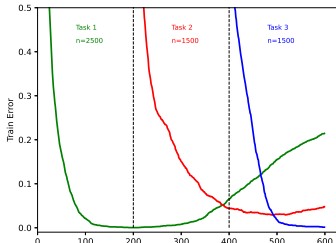 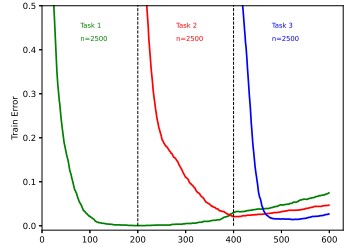 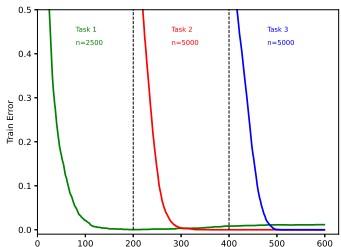

*Figure 2.* Classification train error for each task vs. iterations for the XOR cluster with $K = 3$ tasks trained on a quadratic network. We fix $n = 2500$ for the first task and increase the sample size of the second and third tasks across figures. Increasing the sample size stabilizes per-task training and *decreases forgetting* for *previous tasks*.

## 3. Experiments

We demonstrate the impact of sample size, number of tasks, and network width on the performance of continual learning for different loss functions, activations functions, data distributions, architectures, step-sizes and training horizons. We include the implementation details for each figure and additional experiments, including experiments on the transformer architecture in Appendix E. The code for reproducing the results is publicly available online.[1]

**Impact of sample-size, training horizon and number of Tasks.** The first data model we consider is the XOR cluster with orthogonal mean vectors. Fig. 1 shows how sample-size affects the train-loss forgetting for $K = 3$ tasks using quadratic activation and linear loss. Here, we increase the sample size for each task from $n = 2500$ to $n = 5000$, showing how the increase can diminish test-error forgetting. Fig. 9 in the appendix repeats this experiment for different problem parameters. The observations from both plots are in-line with our theoretical insights on the role of sample-size on train and test time forgetting.

In order to verify the role of sample size of later tasks on train-time forgetting, we consider an experiment where the

---

[1] https://github.com/hosseinta2/continual-learning-with-neural-nets.git

sample-size for task 1 is fixed, and for later tasks we increase the sample-size. The resulting training loss curves for different loss functions and activations are shown in Figs. 2, 3, and 10 (in the appendix). In accordance with Theorem 2.1, it can be observed that increasing the sample-size on tasks 2,3 has a positive influence on the forgetting of task 1. This implies that increasing the sample-size not only stabilizes the per-task training loss, but also reduces the amount of forgetting for previous tasks. While we use linear loss with quadratic activation for Fig. 2, Figs. 3, 10 indicate these observations extend to different losses and activations including the commonly used logistic loss and the ReLU and GELU activations.

In Fig. 4, we consider $K = 6$ tasks of the XOR cluster dataset and increase $T$ from $T = 2000$ to $T = 4000$ for each task with $n = 200, 800, 2000$ samples per each task. Note that increasing $T$, deteriorates the training loss for task 1 as training progresses. While increasing $T$ helps with training loss for task 1 at the end of training of task 1, (the dashed lines are below the solid lines at $k = 1$ for any value of $n$), the amount of increase in the training loss for $T = 4000$ is larger than $T = 2000$, eventually leading to larger training loss for task 1 as $K$ increases. The right panel in Fig. 4 shows the training loss for each task during learning these 6 tasks, illustrating that the train loss achieves near zero training loss for each task. On the other

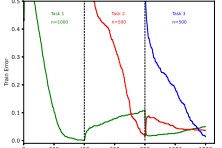 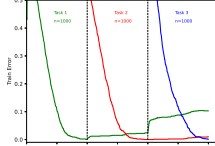 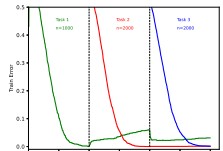 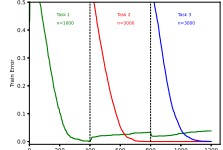 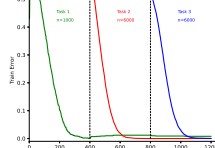

*Figure 3.* We repeat the experiment from Figure 2, this time using *GELU activation* and the *logistic loss function*, demonstrating that our findings remain valid across different settings.

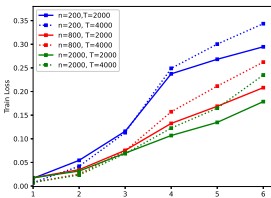 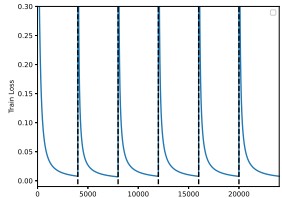

*Figure 4.* Left: Training loss of Task 1 versus task index (i.e., $\widehat{F}_1(w_k)$ as a function of $k$) for $K = 6$ tasks for different sample-sizes and training horizons per task. Right: Training loss per task $(\widehat{F}_k(w_k^{(t)}))$ versus iteration when $n = 2000, T = 4000$ for each task. We use GELU activations with logistic loss. Although each task individually reaches near-zero training loss, the training loss on the first task *increases* with both the *number of tasks* ($K$) and the *training horizon* ($T$), and *decreases* as the sample size ($n$) grows.

hand, increasing $n$ for each task, helps with diminishing the training loss. To better see this impact, in Fig. 11 in the appendix, we increase the number of tasks and consider learning $K = 15$ and $K = 20$ tasks of the XOR cluster dataset. These plots again verify our insights on the role of training-set size. The impact of increasing tasks is also visible in the Left figure while using GELU activation and the logistic loss.

**Impact of over-parameterization.** In Fig. 6 we consider the XOR cluster dataset for $K = 3$ tasks with Quadratic activation and gradually increase $m$ from $m = 10^2$ to $m = 10^4$. We find that increasing the width is generally beneficial for continual learning. However the benefits shrink as $m$ increases, where increasing the width from $m = 10^3$ to $m = 10^4$ has almost non-tangible impact on the overall performance of continual learning. Note that this is in line with Theorem 2.1, as we discussed the impact of width showing that width alone cannot reduce the train time forgetting to zero. We remark these insights also align with the *diminishing returns of width* phenomenon observed in previous works (Guha & Lakshman, 2024; Graldi et al., 2024) where the benefits of width decline as $m$ grows. In Fig. 7, we consider learning $K = 6$ tasks with the GELU activation and logistic loss for different choices of over-parameterization. The observations in this figure again verify our previous insights as increasing the width helps with continual learn-

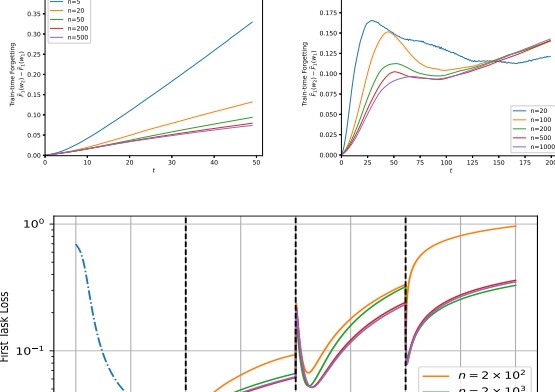

*Figure 5.* Top: Train-time forgetting for task 1 while learning the second task versus iteration for the split-MNIST dataset, classifying labels '0'/'1' for task 1 and labels '2'/'3' for task 2 (left) and labels '4'/'5' for task 1 and labels '6'/'7' for task 2 (right). We fix $n = 50$ samples for the first task, and change $n$ for the second task. Bottom: First task's training loss $(\widehat{F}_1(w^{(t)}))$ vs $t$ for learning 4 binary tasks from the split FashionMNIST dataset. We fix $n = 200$ for Task 1 and plot the training curves while increasing $n$ for subsequent tasks. Similar to our theoretical insights increasing $n$ reduces forgetting for previous tasks.

ing, although it alone cannot lead to forget-less continual learning.

**Experiments with MNIST and FashionMNIST.** In Fig. 5 (Top), we consider continual binary classification of digits from the MNIST dataset with $K = 2$ tasks. The plots show the amount of increase in training loss of task 1, during learning task 2 versus iteration number. The results are averages over 15 independent experiments. For the left plot tasks are determined according to digits 0 to 3 and for the right plot the tasks are determined according to the data distribution formed by digits 4 to 7. The sample-size for the first task is fixed to $n = 50$ in all curves and different curves correspond to different sample sizes for the second task. The results of previous figures on the role of sample-size continue to hold for this distribution as well, since increasing the sample-size for the second task generally improves the continual learning of the first task. In Fig. 5 (Bottom), we

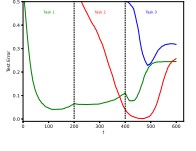 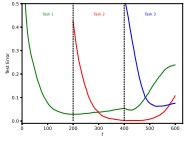 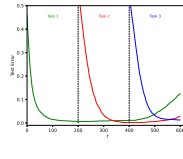 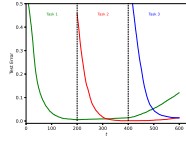 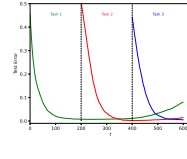 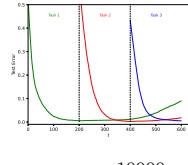

| $m = 100$ | $m = 300$ | $m = 1000$ | $m = 3000$ | $m = 6000$ | $m = 10000$ |

*Figure 6.* Impact of network width ($m$) on the test error for learning the XOR cluster distribution with 3 tasks using quadratic networks. Increasing width helps with continual learning; however, the benefits diminish as $m$ grows, indicating that over-parameterization alone *cannot* eliminate forgetting.

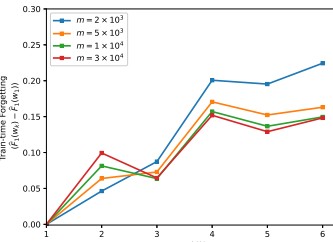

*Figure 7.* Train-time forgetting for task 1 vs task for $K = 6$ total tasks of the XOR cluster dataset for different over-parameterization choices. We use GELU activations with logistic loss and train for $T = 10^3$ iterations per task. The same trend observed in Fig. 6 appears here as well.

consider a similar experiment but with the FashionMNIST dataset, choose logistic loss and ReLU activation, and set the total number of tasks to $K = 4$, where different tasks correspond to data from different labels. Similar to the last experiment, we observe that increasing the sample-size for subsequent tasks generally has a positive impact on the first task's training loss.

## 4. Conclusions and Future Work

We studied gradient-based continual learning in a neural network setup, highlighting how different problem parameters affect catastrophic forgetting. Our analysis provides the first closed-form bounds on train and test time forgetting in this setting and clarifies the roles of sample size, width, number of tasks, and training horizon. There are several promising directions for future work. An immediate next step is to analyze other training methodologies, such as (mini-batch) stochastic gradient descent, where additional noise may interact with forgetting. Another important direction is to move beyond the quadratic two-layer setting and explore whether analogous guarantees can be obtained for richer architectures, including transformers. Our preliminary experiments in Fig. 12 in the appendix show that some aspects of our results are observed, particularly for small transformers with Gaussian-Mixture data. Finally, our current analysis is limited to the lazy regime. Extending the theory to the feature-learning regime, where step-sizes are

large, early stopping is avoided, and weights move significantly from their initialization, remains a challenging and exciting problem. While a recent work (Graldi et al., 2024) provides preliminary results on the drawbacks of feature learning for continual learning, more exploration in this regime could provide a more complete picture of continual learning in modern machine learning.

**Acknowledgment.** This work was supported by NSF awards 2217058 and 2112665.

## Impact Statement

This paper presents work whose goal is to advance the field of machine learning. There are many potential societal consequences of our work, none of which we feel must be specifically highlighted here.

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

# Appendix

## A. Single-task XOR Cluster

The next result derives the class margin for the single-task XOR cluster dataset and combined with standard results from the NTK literature it bounds the train and test loss for learning this dataset (as described by Eq. 1) with GD.

**Proposition A.1** (Single-task XOR). *For the XOR cluster dataset for a given $T$, there exists target vector $w^\star \in \mathbb{R}^{dm}$ such that $\|w^\star - w_0\| = \Theta(d \cdot \log(T))$ and $\widehat{F}(w^\star) < 1/T$ and gradient descent with logistic loss and on a network with quadratic activation with width $m = \Omega(\|w^\star - w_0\|^4)$, achieves the training loss $\widehat{F}(w_t) = O(\frac{\|w^\star - w_0\|^2}{t})$ and the expected test loss $\mathbb{E}_{\mathcal{D}}[F(w_t)] = O(\frac{\|w^\star - w_0\|^2}{n})$ after $t = n$ GD iterations.*

*Proof.* Define four regions $R_1, R_2, R_3, R_4 \in \mathbb{R}^d$ such that

$$
\begin{aligned}
R_1 &= \{x \in \mathbb{R}^d : x^\top(\mu_+ + \mu_-) > 0, x^\top(\mu_+ - \mu_-) > 0\}, \\
R_2 &= \{x \in \mathbb{R}^d : x^\top(\mu_+ + \mu_-) > 0, x^\top(\mu_+ - \mu_-) < 0\}, \\
R_3 &= \{x \in \mathbb{R}^d : x^\top(\mu_+ + \mu_-) < 0, x^\top(\mu_+ - \mu_-) > 0\}, \\
R_4 &= \{x \in \mathbb{R}^d : x^\top(\mu_+ + \mu_-) < 0, x^\top(\mu_+ - \mu_-) < 0\}.
\end{aligned}
$$

Without loss of generality, assume $\mu_+ = [1/\sqrt{d}, 1/\sqrt{d}, 0, \cdots, 0]$ and $\mu_- = [-1/\sqrt{d}, 1/\sqrt{d}, 0 \cdots, 0]$. Our goal is to derive the NTK margin (Ji & Telgarsky, 2020; Taheri & Thrampoulidis, 2024) denoted by $\gamma$ for infinitely wide neural networks with initialization variable $z \in \mathbb{R}^d$, i.e., show that the equation below holds for all data points in the training set almost surely:

$$
M(x_i, y_i) := y_i \int_{z \in \mathbb{R}^d} \phi'(\langle z, x_i \rangle) \langle w_z, x_i \rangle \, d\mu_{\mathcal{N}}(z) \geq \gamma
$$

where $\mu_N$ is the standard Gaussian measure and $w_z$ is an initialization dependent vector such that $\|w_z\| \leq 1$ for all $z \in \mathbb{R}^d$. We drop the subscript $i$ and assume quadratic activation. Assume $y = 1, x \sim \mathcal{N}(\mu_+, \sigma I_d)$ without loss of generality. Then,

$$
\begin{aligned}
M = &\int_{z \in R_1} \langle z, x \rangle \langle w_z, x \rangle \, d\mu_{\mathcal{N}}(z) + \int_{z \in R_2} \langle z, x \rangle \langle w_z, x \rangle \, d\mu_{\mathcal{N}}(z) \\
&+ \int_{z \in R_3} \langle z, x \rangle \langle w_z, x \rangle \, d\mu_{\mathcal{N}}(z) + \int_{z \in R_4} \langle z, x \rangle \langle w_z, x \rangle \, d\mu_{\mathcal{N}}(z)
\end{aligned}
$$

Let

$$
w_z = \mu_+/\|\mu_+\|, -\mu_-/\|\mu_-\|, \mu_-/\|\mu_-\|, -\mu_+/\|\mu_+\| \text{ if } z \in R_1, R_2, R_3, R_4, \text{ respectively.}
$$

Assume $s \sim \mathcal{N}(0, \sigma I_d)$.

$$
\begin{aligned}
&\langle \mu_+/\|\mu_+\|, \mu_+ + s \rangle \int_{z \in R_1} \langle z, \mu_+ + s \rangle \, \mathrm{d}\mu_{\mathcal{N}}(z) \\
&= (\|\mu_+\| + \mu_+^\top s/\|\mu_+\|) \left( (\frac{1}{\sqrt{d}} + s(1)) \int_{z \in R_1} z(1) d\mu_{\mathcal{N}}(z) + (\frac{1}{\sqrt{d}} + s(2)) \int_{z \in R_1} z(2) d\mu_{\mathcal{N}}(z) \right) \\
&= (\|\mu_+\| + \frac{\mu_+^\top s}{\|\mu_+\|})(\frac{2}{\sqrt{d}} + s(1) + s(2)) \mathbb{E}[z(1)_{z(1)>0}] \\
&= \left( \sqrt{\frac{2}{d}} + \frac{\sqrt{2}}{2}(s(1) + s(2)) \right) \left( \frac{2}{\sqrt{d}} + s(1) + s(2) \right) \frac{1}{\sqrt{2}} \\
&\gtrsim (\frac{1}{\sqrt{d}} + s(1) + s(2))^2 \\
&\gtrsim (\frac{1}{\sqrt{d}} + O(\frac{1}{\sqrt{d} \cdot \log^c(d)}))^2 \\
&\gtrsim 1/d.
\end{aligned}
$$

For the second integral we have,

$$\langle -\mu_- / \|\mu_-\|, \mu_+ + s\rangle \int_{z \in R_2} \langle z, \mu_+ + s\rangle \, \mathrm{d}\mu_\mathcal{N}(z)$$

$$= \frac{-\mu_-^\top s}{\|\mu_-\|} \left( (\frac{1}{\sqrt{d}} + s(1)) \int_{z \in R_2} z(1) \, \mathrm{d}\mu_\mathcal{N}(z) + (\frac{1}{\sqrt{d}} + s(2)) \int_{z \in R_2} z(2) \, \mathrm{d}\mu_\mathcal{N}(z) \right)$$

$$= \frac{-1}{2}(s(2) - s(1))^2 = \Theta(\frac{1}{d \cdot \log^{2c}(d)})$$

For the third and fourth integral, due to symmetry, we reach the above final results again. Overall, we find that

$$M(x_i, y_i) \gtrsim \frac{1}{d} + O(\frac{1}{d \cdot \mathrm{poly}\log(d)}) = \Omega(1/d).$$

For data points coming from other three clusters of the XOR distribution, we reach the same conclusion. Therefore the margin scales as $1/d$ for every training sample. Using this margin result in (Taheri & Thrampoulidis, 2024, Corollary C.1.1 and Proposition C.1) completes the result. □

## B. Proofs for Bounds on Delayed Generalization Gap: Proof of Theorem 2.3

We state and prove a more general version of Theorem 2.3 in Theorem B.1 below, which requires smaller network width under additional assumptions. We then show that Theorem 2.3 follows as a consequence. Specifically, under additional conditions on continual learnability of each task and a self-bounded assumption for the loss function(i.e., $|f'(u)| < f(u)$) that includes logistic loss $f(u) = \log(1 + \exp(-u))$, in the next theorem, we prove a tight generalization bound, which has a noticeably milder dependence on $T$ compared to Theorem 2.3.

**Theorem B.1** (Improved gen. gap). *Assume the loss function is self-bounded, 1-Lipschitz and 1-smooth. Let the network's width $m$ be large enough so that $\sqrt{m} \gtrsim \eta \sum_{j=k+1}^{K} \sum_{t=0}^{T-1} \widehat{F}_j(w_j^{(t)})$. Moreover, assume there exists $w_k^\star$ achieving small training loss $\widehat{F}_k(w_k^\star) \leq \|w_k^\star - w_k^{(0)}\|^2/(\eta T)$ for task $k$, and satisfying $m \gtrsim \|w_k^\star - w_k^{(0)}\|^4$. Then,*

$$\mathcal{F}_{k,K}^{\mathrm{gen}} \lesssim \frac{\eta}{n} \mathbb{E}_{\mathcal{D}_k} \left[ e^{\frac{\eta}{\sqrt{m}} c_{k,K}} \sum_{t=0}^{T-1} \widehat{F}_k(w_k^{(t)}) \right], \tag{6}$$

*where $c_{k,K} = O\left( \sum_{j=k+1}^{K} \sum_{t=0}^{T-1} \widehat{F}_j(w_j^{(t)}) \right)$.*

As the result shows, $\mathcal{F}_{k,K}^{\mathrm{gen}}$ decays with both the cumulative training loss of the later tasks as in $c_{k,K}$ and the network width, and it is proportional to the cumulative training loss of task $k$. In particular, the cumulative training loss can be much smaller than $T$, potentially leading to tighter bounds compared to the results of previous theorem.

*Remark* B.2. In words, the conditions on $\|w_k^\star - w_k^{(0)}\|$ ensure that task $k$ remains learnable in the kernel regime, i.e., the initialization is sufficiently close to the task-specific optimum so that optimization can succeed. To better interpret this result, let us consider the case where $k = 1$, and we are interested in bounds on $\mathcal{F}_{1,K}^{\mathrm{gen}}$ for some $K \geq 2$. First, we note that for the XOR cluster dataset, there exists(see Proposition A.1 in App. A) $w_1^\star$ such that $\|w_1^\star - w_1^{(0)}\| = \Theta(d \cdot \log(T))$ and $\widehat{F}_1(w_1^\star) \leq \frac{1}{T}$, leading to train-loss $\widehat{F}_1(w_1^{(t)}) = O(\frac{d^2 \log^2(t)}{t})$. Therefore, in view of Theorem B.1, if $\sqrt{m} \gtrsim \eta \sum_{j=2}^{K} \sum_{t=0}^{T-1} \widehat{F}_j(w_j^{(t)})$ and $m \gtrsim d^4 \log^4(T)$, the expected generalization gap after $T$ iterations for each of $K$ tasks satisfies,

$$\mathcal{F}_{1,K}^{\mathrm{gen}} \lesssim \frac{\eta d^2 \log^3(T)}{n},$$

where we can hide the exponential term in Eq. 6 for simplicity since the exponent is constant under the condition on $m$. This shows that Theorem B.1 may lead to bounds with significantly better dependence based on $T$ compared to Theorem 2.3 (poly-logarithmic vs linear). Although this result cannot be combined directly with our setting for the training loss (since the hinge loss considered for the training-loss analysis is not self-bounded) it still provides valuable insight. In particular, it can be interpreted as a stronger extension of Theorem 2.3, highlighting how the training loss directly influences the generalization gap in continual learning as shown by Eq. 6.

## B.1. Proof of Theorem B.1

Recall the continual learning of $K$ tasks for $T$ iterations each i.e., at task $k \in [K]$ :

$$w_t = w_{t-1} - \eta \nabla \widehat{F}_k(w_t) \text{ for } (k-1) \cdot T < t \le kT$$

Assume $f(\cdot, x)$ to be the sample loss which is $L$-Lipschitz with respect to its first input. Let $\mathcal{D}_k := \{x_1, \cdots, x_n\}$ be the training dataset of task $k$. Denote $w_\ell^{\neg i}$ as the output of the continual learning algorithm after learning task $\ell$ (for some $\ell \le K$), when $x_i$ is left out of the training samples from $\mathcal{D}_k$. Similarly, we define $w_\ell^{(t), \neg i}$, as the output of continual learning at iteration $t$ of task $\ell$ when $x_i$ is left out.

The generalization gap associated with task $k$, after learning $K$ tasks can be written as,

$$\mathbb{E}_{\mathcal{D}_k}[F_k(w_K) - \widehat{F}_k(w_K)] = \frac{1}{n} \sum_{i=1}^n \mathbb{E}_{\mathcal{D}_k, x} \left[ f(w_K, x) - f(w_K, x_i) \right]$$

$$= \frac{1}{n} \sum_{i=1}^n \mathbb{E}_{\mathcal{D}_k, x} \left[ f(w_K, x) - f(w_K^{\neg i}, x_i) \right] + \frac{1}{n} \sum_{i=1}^n \mathbb{E}_{\mathcal{D}_k} \left[ f(w_K^{\neg i}, x_i) - f(w_K, x_i) \right]$$

$$= \frac{1}{n} \sum_{i=1}^n \mathbb{E}_{\mathcal{D}_k, x} \left[ f(w_K, x) - f(w_K^{\neg i}, x) \right] + \frac{1}{n} \sum_{i=1}^n \mathbb{E}_{\mathcal{D}_k} \left[ f(w_K^{\neg i}, x_i) - f(w_K, x_i) \right]$$

$$\le \frac{L}{n} \sum_{i=1}^n \mathbb{E}_{\mathcal{D}_k} \left[ \| w_K - w_K^{\neg i} \| \right]. \tag{7}$$

Therefore, the samples from the subsequent distributions do not impact the delayed generalization gap in Task $k$, as we are taking the expectation over only $\mathcal{D}_k$.

$$\| w_K - w_K^{\neg i} \| = \| w_K^{(T-1)} - \eta \nabla \hat{F}_K(w_K^{(T-1)}) - (w_K^{(T-1), \neg i} - \eta \nabla \hat{F}_K(w_K^{(T-1), \neg i})) \|$$

Note that the objectives are the same for both $w_K^{(T-1), \neg i}$ and $w_K^{(T-1)}$. Therefore, by the non-expansive properties of one-hidden-layer neural nets (Taheri & Thrampoulidis, 2024, Lemma B.1):

$$\| w_K - w_K^{\neg i} \| \le \left( 1 + \frac{\eta L R^2}{\sqrt{m}} \max_{w_\alpha \in [w_K^{(T-1)}, w_K^{(T-1), \neg i}]} \widehat{F}'_K(w_\alpha) \right) \| w_K^{(T-1)} - w_K^{(T-1), \neg i} \| \tag{8}$$

where we define:

$$\widehat{F}'(w) := \frac{1}{n} \sum_{i=1}^n |f'(w, x_i)|,$$

with $f'$ denoting the derivative of the sample loss. For self-bounded losses assumed in this theorem, we have $|f'(w, x_i)| \le f(w, x_i)$, therefore $\widehat{F}'_K(w_\alpha) \le \widehat{F}_K(w_\alpha)$, leading to:

$$\| w_K - w_K^{\neg i} \| \le \left( 1 + \frac{\eta L R^2}{\sqrt{m}} \max_{w_\alpha \in [w_K^{(T-1)}, w_K^{(T-1), \neg i}]} \widehat{F}_K(w_\alpha) \right) \| w_K^{(T-1)} - w_K^{(T-1), \neg i} \|$$

Repeating this step for $T$ steps from $T$ to 1:

$$\| w_K - w_K^{\neg i} \| \le \prod_{t=0}^{T-1} \left( 1 + \frac{\eta L R^2}{\sqrt{m}} \max_{w_{\alpha t} \in [w_K^{(t)}, w_K^{(t), \neg i}]} \widehat{F}_K(w_{\alpha t}) \right) \left\| w_K^{(0)} - w_K^{(0), \neg i} \right\|$$

$$= \prod_{t=0}^{T-1} \left( 1 + \frac{\eta L R^2}{\sqrt{m}} \max_{w_{\alpha t} \in [w_{K-1}^{(t)}, w_K^{(t), \neg i}]} \widehat{F}_K(w_{\alpha t}) \right) \left\| w_K - w_K^{\neg i} \right\|$$

$$\le \exp \left( \frac{\eta L R^2}{\sqrt{m}} \sum_{t=0}^{T-1} \max_{w_{\alpha t} \in [w_K^{(t)}, w_K^{(t), \neg i}]} \widehat{F}_K(w_{\alpha t}) \right) \left\| w_K - w_K^{\neg i} \right\|.$$

where $R$ is the max norm of data and $L$ is the activation function's Lipschitz parameter. We need an inductive argument here to prove that $\|w_t - w_t^{\neg i}\|$ remains bounded for all $t$ as it is used in the max over $w_{\alpha t}$ term.

Repeating this step for $K - k$ tasks, we derive the following,

$$\|w_K - w_K^{\neg i}\| \le \exp\left(\frac{\eta L R^2}{\sqrt{m}} \sum_{j=k+1}^{K} \sum_{t=0}^{T-1} \max_{w_{\alpha t} \in [w_j^{(t)}, w_j^{(t), \neg i}]} \widehat{F}_j(w_{\alpha t})\right) \|w_k - w_k^{\neg i}\|. \tag{9}$$

This gives an expression for bounding the generalization gap based on the parameter stability of the $k'$th task, the width and training performance from task $k + 1$ to task $K$. To bound the parameter stability term, note that,

$$\begin{aligned}
\|w_k - w_k^{\neg i}\| &\le \left\|w_k^{(T-1)} - \eta \nabla \widehat{F}_k(w_k^{(T-1)}) - (w_k^{(T-1), \neg i} - \eta \nabla \widehat{F}_k(w_k^{(T-1), \neg i}))\right\| \\
&\quad + \eta \left\|\nabla \widehat{F}_k^i(w_k^{(T-1), \neg i})\right\|
\end{aligned}$$

Recall the $i$th data point is taken from the $k$th task data distribution. For tasks $j$ where $j < k$, it holds that $w_j = w_j^{\neg i}$. Therefore we can use the result from previous works (Taheri & Thrampoulidis, 2024, Thm B.2) on the stability error of neural networks in the NTK regime to bound $\left\|w_k - w_k^{\neg i}\right\|$.

**Lemma B.3.** *If there exists $w_k^\star$ such that $\|w_k^\star - w_k\| \ge \max\left\{\sqrt{\eta T \widehat{F}_k(w_k^\star)}, \sqrt{\eta \widehat{F}_k(w_{k-1})}\right\}$, and $m \gtrsim \|w_k^\star - w_k\|^4$, then $\left\|w_k - w_k^{\neg i}\right\| \lesssim \frac{\eta}{n} \sum_{t=0}^{T-1} \widehat{F}_k^i(w_k^{(t)})$, and consequently,*

$$\mathbb{E}_{\mathcal{D}_k}\left[\frac{1}{n} \sum_{i=1}^{n} \left\|w_k - w_k^{\neg i}\right\|\right] \lesssim \frac{\eta}{n} \sum_{t=0}^{T-1} \mathbb{E}_{\mathcal{D}_k}\left[\widehat{F}_k(w_k^{(t)})\right].$$

Let us define $c_{k,K} := \max_{i \in [n]} \sum_{j=k+1}^{K} \sum_{t=0}^{T-1} \max_{w_{\alpha t} \in [w_j^{(t)}, w_j^{(t), \neg i}]} \widehat{F}_j(w_{\alpha t})$. Then by this lemma we have,

$$\mathbb{E}_{\mathcal{D}_k}\left[\frac{1}{n} \sum_{i=1}^{n} \|w_K - w_K^{\neg i}\|\right] \le \frac{\eta \, e^{\frac{\eta}{\sqrt{m}} c_{k,K}}}{n} \sum_{t=0}^{T-1} \mathbb{E}_{\mathcal{D}_k}\left[\widehat{F}_k(w_k^{(t)})\right].$$

### B.1.1. BOUNDING $c_{k,K}$

In order to bound $c_{k,K}$, we use the following result on the quasi-convexity properties of the two-layer neural net objective by (Taheri & Thrampoulidis, 2024, Prop. 5.1.).

**Lemma B.4.** *Suppose $\widehat{F} : \mathbb{R}^{d'} \to \mathbb{R}$ satisfies the self-bounded weak convexity property with parameter $\kappa$. Let $w_1, w_2 \in \mathbb{R}^{d'}$ be two arbitrary points with distance $\|w_1 - w_2\| \le D < \sqrt{2/\kappa}$. Set $\tau := \left(1 - \kappa D^2/2\right)^{-1}$. Then,*

$$\max_{v \in [w_1, w_2]} \widehat{F}(v) \le \tau \cdot \max\left\{\widehat{F}(w_1), \widehat{F}(w_2)\right\}.$$

For self-bounded losses $\kappa = \frac{1}{\sqrt{m}}$, therefore if $w, w'$ are such that $\|w - w'\| \le D \le m^{1/4}$, then

$$\max_{v \in [w, w']} \widehat{F}(v) \le \frac{1}{1 - \frac{D^2}{\sqrt{m}}} \cdot \max\left\{\widehat{F}(w), \widehat{F}(w')\right\}.$$

Recall,

$$\|w_K - w_K^{\neg i}\| \le \exp\left(\frac{\eta}{\sqrt{m}} \sum_{j=k+1}^{K} \sum_{t=0}^{T-1} \max_{w_{\alpha t} \in [w_j^{(t)}, w_j^{(t), \neg i}]} \widehat{F}_j(w_{\alpha t})\right) \|w_k - w_k^{\neg i}\|.$$

Assume

$$\sqrt{m} \geq 8 \max \left\{ \eta \sum_{j=k+1}^{K} \sum_{t=0}^{T-1} (\widehat{F}_j(w_j^{(t)}) + \widehat{F}_j(w_j^{(t),\neg i})), \|w_k - w_k^{\neg i}\|^2 \right\}. \tag{10}$$

Then, by induction $\|w_j^{(t)} - w_j^{(t),\neg i}\| \leq 2\|w_k - w_k^{\neg i}\|$ for all $t \in [T], j \in [k, K]$. To see this:

$$\|w_j^{(t)} - w_j^{(t),\neg i}\|$$
$$\leq \exp \left( \frac{\eta}{\sqrt{m}} \sum_{j'=k+1}^{j-1} \sum_{\tau=0}^{T-1} \max_{w_{\alpha\tau} \in [w_{j'}^{(\tau)}, w_{j'}^{(\tau),\neg i}]} \widehat{F}_{j'}(w_{\alpha\tau}) + \sum_{\tau=0}^{t-1} \max_{w_{\alpha\tau} \in [w_j^{(\tau)}, w_j^{(\tau),\neg i}]} \widehat{F}_j(w_{\alpha\tau}) \right)$$
$$\times \|w_k - w_k^{\neg i}\|$$

By induction's assumption $\sqrt{m} \geq 2\|w_{j'}^{(\tau)} - w_{j'}^{(\tau),\neg i}\|^2$. Therefore we can invoke Lemma B.4 for all the $\max \widehat{F}_{j'}$ to find that,

$$\|w_j^{(t)} - w_j^{(t),\neg i}\| \leq \exp(1/4) \cdot \|w_k - w_k^{\neg i}\| \leq 2 \|w_k - w_k^{\neg i}\|.$$

Which proves the induction. Overall, we could bound $c_{K,k}$ based on the training objective. assuming $\widehat{F}_j(w_j^{(t)})$ and $\widehat{F}_j(w_j^{(t),\neg i})$ are of the same order(needs proof), then we find

$$c_{K,k} \leq 2 \sum_{j=k+1}^{K} \sum_{t=0}^{T-1} (\widehat{F}_j(w_j^{(t)}) + \widehat{F}_j(w_j^{(t),\neg i})) = O \left( \sum_{j=k+1}^{K} \sum_{t=0}^{T-1} \widehat{F}_j(w_j^{(t)}) \right)$$

To simplify the statement of the lemma, we can assume $\widehat{F}_j(w_j^{(t)})$ and $\widehat{F}_j(w_j^{(t),\neg i})$ are of the same order as reducing the sample-size by 1 sample does not affect the training bounds.

**Lemma B.5.** *Let the assumptions of Lemma B.3 hold. Assume*

$$\sqrt{m} \gtrsim \eta \sum_{j=k+1}^{K} \sum_{t=0}^{T-1} (\widehat{F}_j(w_j^{(t)}) + \widehat{F}_j(w_j^{(t),\neg i})) \asymp \eta \sum_{j=k+1}^{K} \sum_{t=0}^{T-1} \widehat{F}_j(w_j^{(t)}).$$

*Then,*

$$\mathbb{E}_{\mathcal{D}_k} \left[ \frac{1}{n} \sum_{i=1}^{n} \|w_K - w_K^{\neg i}\| \right] \leq \frac{\eta}{n} \mathbb{E}_{\mathcal{D}_k} \left[ e^{\frac{\eta}{\sqrt{m}} c_{k,K}} \sum_{t=0}^{T-1} \widehat{F}_k(w_k^{(t)}) \right]$$

*where $c_{k,K} = O\left( \sum_{j=k+1}^{K} \sum_{t=0}^{T-1} \widehat{F}_j(w_j^{(t)}) \right)$.*

*Proof.* The proof essentially follows by the last two lemmas and noting that $\|w_k - w_k^{\neg i}\| \leq \|w_k - w_{k-1}\| + \|w_k^{\neg i} - w_{k-1}\| = O(\|w_k^\star - w_{k-1}\|)$ by Lemma B.3. Therefore the condition we had in Eq. 10 on $\sqrt{m} \geq \|w_k - w_k^{\neg i}\|^2$ is absorbed in the condition from Lemma B.3.

$\square$

This completes the proof of the Theorem. Having established Theorem B.1, we are now ready to prove our main result on generalization in Theorem 2.3.

## B.2. Proof of Theorem 2.3

**Theorem B.6** (Restatement of Theorem 2.3). *Assume the loss function is 1-Lipschitz and 1-smooth. Then, the expected delayed generalization gap satisfies,*

$$\mathcal{F}_{k,K}^{\text{gen}} := \mathbb{E}_{\mathcal{D}_k}\left[F_k(w_K) - \widehat{F}_k(w_K)\right] \lesssim \frac{\eta T\, e^{\frac{\eta T(K-k+1)}{\sqrt{m}}}}{n}.$$

*Proof.* The proof of Theorem 2.3 essentially follows from Theorem B.1. We outline the distinct steps. Note that since the objective is 1-Lipschitz, it holds $\widehat{F}'(w) \leq 1$ for any $w$.. Therefore Eq. 8 from the proof of Theorem 2.3 changes into

$$\|w_K - w_K^{\neg i}\| \leq \left(1 + \frac{\eta L R^2}{\sqrt{m}}\right)\left\|w_K^{(T-1)} - w_K^{(T-1),\neg i}\right\|.$$

As a result, by unrolling the iterates and noting that $R \leq 1$:

$$\left\|w_K - w_K^{\neg i}\right\| \leq \exp\left(\frac{\eta L(K-k)T}{\sqrt{m}}\right)\left\|w_k - w_k^{\neg i}\right\|. \tag{11}$$

Moreover, again using the Lipschitz loss function properties:

$$\begin{aligned}
\left\|w_k - w_k^{\neg i}\right\| &\leq \left\|w_k^{(T-1)} - \eta\nabla\widehat{F}_k(w_k^{(T-1)}) - (w_k^{(T-1),\neg i} - \eta\nabla\widehat{F}_k(w_k^{(T-1),\neg i}))\right\| \\
&\quad + \eta\left\|\nabla\widehat{F}_k^i(w_k^{(T-1),\neg i})\right\| \\
&\leq \left\|w_k^{(T-1)} - \eta\nabla\widehat{F}_k(w_k^{(T-1)}) - (w_k^{(T-1),\neg i} - \eta\nabla\widehat{F}_k(w_k^{(T-1),\neg i}))\right\| + \frac{\eta L}{n} \\
&\leq \exp(\frac{\eta L}{\sqrt{m}})\|w_k^{(T-1)} - w_k^{(T-1),\neg i}\| + \frac{\eta L}{n} \\
&\leq \exp(\frac{2\eta L}{\sqrt{m}})\|w_k^{(T-2)} - w_k^{(T-2),\neg i}\| + (1 + \exp(\frac{\eta L}{\sqrt{m}}))\frac{\eta L}{n} \\
&\leq (\sum_{t=0}^{T-1}\exp(\frac{\eta L t}{\sqrt{m}}))\frac{\eta L}{n} \\
&\leq \exp(\frac{\eta L T}{\sqrt{m}})\frac{\eta L T}{n}.
\end{aligned}$$

where the last step is derived by repeating the procedure over all $T$ iterations.

Inserting this in Eq. 11, taking the expectation over $\mathcal{D}_k$, using Eq. 7 and noting that $L$ (the objective's Lipschitz parameter) is constant for our setup, conclude the proof of the theorem. $\qquad\square$

# C. Bounding Train-time Loss and Forgetting for XOR Cluster Data

In this section, we prove Theorems 2.1-2.2. Below, is a restatement of these theorems.

**Theorem C.1** (Restatement of Theorems 2.1-2.2). *Consider the $d$-dimensional XOR cluster dataset with $K$ tasks and assume gradient descent with $\eta T = \Theta(d^2)$ iterations and $n = \widetilde{\Theta}(d^2 K)$ samples for each subsequent task trained by a neural net with $m = \widetilde{\Omega}(d^8 K^4)$ hidden neurons. Then, with high probability, the train-time forgetting and per-task train-time time error is $\mathcal{F}_{k,K}^{\text{tr}} = o_d(1)$. In particular, for the train-time forgetting with probability $1 - \delta$, we have:*

$$|\mathcal{F}_{k,K}^{\text{tr}}| := |\widehat{F}_k(w_K) - \widehat{F}_k(w_k)| = \widetilde{O}\left(\eta T\frac{\sqrt{K-k}}{d\sqrt{n}} + \eta T\frac{\sqrt{K-k}}{d^2\operatorname{poly}\log(d)} + \eta^2 T^2\frac{K^2}{\sqrt{m}}\right),$$

*where $\widetilde{O}(\cdot)$ hides logarithmic factors in $n$ and $\delta$.*

The proof strategy is as follows. First, we consider the $m \to \infty$ and derive the weights for arbitraay number of GD steps for each task. We then show that for sufficiently large $T$ and sufficiently large $n$, and by computing the network output via concentration bounds based on $n$ for the considered XOR cluster dataset, the train-loss and forgetting are approximately zero. We then compute the error due to finite-width, showing that under sufficiently small $T$, and sufficiently large $m$, the derivations of the infinite-width regime are approximately correct. This leads to the desired quantities and train-time forgetting bounds based $n, T$ and $m$ as stated in the theorem. We start by considering the infinite width regime.

## C.1. Training Error of an Infinitely Wide Network

First, we consider the $m \to \infty$ regime and characterize the distribution of the final weights after $T$ and $2T$ iterations in this regime. We then discuss the general formula for arbitrary number of tasks. Recall, we considered the hinge-loss for training-time analysis. However, as mentioned in the main body of the paper and as it will become clear in the following analysis, we can simplify the arguments by noting that throughout the optimization process for all $K$ tasks, only the linear part of the loss is used. Thus we can assume the loss function as $f(u) = 1 - u$ without loss of generality.

Let us simplify the notation by droping the task index from weights and instead denoting the vector entering the $i$th neuron by $w^i \in \mathbb{R}^d$. Note that by Taylor expansion around the Gaussian initialization $w_9$, we have,

$$\Phi(w, x) = \Phi(w_0, x) + \frac{1}{\sqrt{m}} \sum_{i=1}^{m} a_i \phi'(\langle w_0^i, x \rangle) \langle x, w^i - w_0^i \rangle + O(\frac{\|w - w_0\|}{\sqrt{m}}).$$

For $w$ close to $w_0$, and for large enough $m$ we can use a linearized neural network model. In particular, in the $m \to \infty$ regime, the updates of the continual learning algorithm are the following for sufficiently small $T$:

$$w_1^i = w_0^i + \eta \frac{1}{\sqrt{m}} \frac{1}{n} \sum_{j=1}^{n} a_i \phi'(\langle w_0^i, x_j^1 \rangle) x_j^1 y_j^1$$

$$w_T^i = w_0^i + \frac{\eta T}{\sqrt{m}} \frac{1}{n} \sum_{j=1}^{n} a_i \phi'(\langle w_0^i, x_j^1 \rangle) x_j^1 y_j^1$$

$$w_{2T}^i = w_0^i + \frac{\eta T}{\sqrt{m}} \frac{1}{n} \sum_{j=1}^{n} a_i \phi'(\langle w_0^i, x_j^1 \rangle) x_j^1 y_j^1 + \frac{\eta T}{\sqrt{m}} \frac{1}{n} \sum_{j=1}^{n} a_i \phi'(\langle w_0^i, x_j^2 \rangle) x_j^2 y_j^2$$

We consider $x_j^k, y_j^k$ for any $j \in [n]$ and $k \in [K]$ as fixed training points used for training task $k$. We consider randomness only with respect to the initialization $w_0^i$ and characterize the distribution of weights in the infinite width regime. As $m \to \infty$ given the IID initialization for $w_0^i$ and the quadratic activation, we deduce the following convergence in distribution,

$$w_T^i = w_0^i + \frac{\eta T}{\sqrt{m}} \frac{1}{n} \sum_{j=1}^{n} a_i \phi'(\langle w_0^i, x_j \rangle) x_j^1 y_j^1 \to \omega z + \frac{\eta t}{\sqrt{m}} \frac{1}{n} \sum_{j=1}^{n} z^\top x_j^1 \, x_j^1 y_j^1 \tag{12}$$

where $\omega \in \{\pm 1\}, z \in \mathbb{R}^d$ are Rademacher random variable and standard Gaussian random vector, respectively, and they represent first layer and second layer initialization.

Let us briefly consider the matrix formulation,

$$R = \frac{1}{n} \sum_{j=1}^{n} y_j^1 x_j^1 z^\top x_j^1 =: Az$$

then $R \sim \mathcal{N}(0, A^2)$ as $Cov(R) = \mathbb{E}[Azz^\top A^\top] = A\mathbb{E}[zz^\top]A^\top = AA^\top = A^2$. In the infinite $n$ asymptotic, $A \to \mathbb{E}[y_j x_j x_j^\top] = \frac{1}{2}\mathbb{E}[xx^\top | y = 1] - \frac{1}{2}\mathbb{E}[xx^\top | y = -1] = \frac{1}{2}(\mu_+^1 \mu_+^{1\top} - \mu_-^1 \mu_-^{1\top})$, indicating that the GD updates learn the true vectors in the $n \to \infty$ regime.

A similar argument leads to the following update rule for the second task:

$$w_{2T}^i \sim z + \frac{\eta T}{\sqrt{m}} A_1 \omega z + \frac{\eta T}{\sqrt{m}} A_2 \omega z, \ \ A_1 := \frac{1}{n} \sum_{j=1}^n y_j^1 x_j^1 {x_j^1}^\top, \ \ A_2 := \frac{1}{n} \sum_{j=1}^n y_j^2 x_j^2 {x_j^2}^\top$$

where again $z \sim \mathcal{N}(0, I_d)$ and $\omega$ is a Rademacher r.v. for representing the binary second layer weights $a_i$.

Similarly, we find that after $K$ tasks with $T$ iterations for each task, the weight $w_{KT}^i$ takes the following form:

$$w_{KT}^i \sim z + \frac{\eta T}{\sqrt{m}} \sum_{j=1}^K A_j \omega z, \ \ A_j := \frac{1}{n} \sum_{v=1}^n y_v^j x_v^j {x_v^j}^\top$$

Recalling the expression for the neural network output, we can characterize the output of the network with this random variable in the infinitely wide regime:

$$\Phi(w_{KT}, x) = \frac{1}{\sqrt{m}} \sum_{i=1}^m a_i \left( \langle w_{KT}^i, x \rangle \right)^2 \sim \frac{1}{\sqrt{m}} \sum_{i=1}^m \omega_i \left( \left\langle z_i + \frac{\eta T}{\sqrt{m}} \sum_{j=1}^K A_j \omega_i z_i, x \right\rangle \right)^2$$

$$= \frac{\eta T}{m} \sum_{i=1}^m \langle z_i, x \rangle \left\langle (\sum_{j=1}^K A_j) z_i, x \right\rangle + \frac{1}{\sqrt{m}} \sum_{i=1}^m \omega_i (z_i^\top x)^2$$

$$+ \frac{\eta^2 T^2}{m\sqrt{m}} \sum_{i=1}^m \omega_i \left( \left\langle z_i + \frac{\eta T}{\sqrt{m}} \sum_{j=1}^K A_j \omega_i z_i, x \right\rangle \right)^2$$

when $m \to \infty$:

$$\longrightarrow \eta T \, \mathbb{E}_z \left[ \langle z, x \rangle \left\langle (\sum_{j=1}^K A_j) z, x \right\rangle \right] + N + 0$$

$$= \eta T \, x^\top (\sum_{j=1}^K A_j) x + N \tag{13}$$

where the last step is by the law of large number and $N$ denotes the asymptotic distribution of the second term. The last term vanishes by the law of large numbers. We derive the training loss by calculating the above for $x$ coming from the training distribution.

We discuss the role of each term in Eq. 13. First, considering the first term above, the training loss for task $K$ w.r.t the first training sample is the following,

$$\frac{\eta T}{n} {x_1^K}^\top \sum_{k=1}^K \sum_{i=1}^n y_i^k x_i^k {x_i^k}^\top x_1^K \tag{14}$$

We split the summation into the relevant task $k = K$ and other tasks when $k \neq K$.

**Case I: $k = K$.** Let us drop $K$ in Eq. 14. we have

$$\frac{1}{n} x_1^\top \sum_{i=1}^n y_i x_i x_i^\top x_1 = \frac{1}{n} \sum_{i=1}^n y_i (x_i^\top x_1)^2 = \frac{1}{n} \left( y_1 \|x_1\|^4 + \sum_{i=2}^n y_i (x_i^\top x_1)^2 \right).$$

Recall our data model:

$$x \sim \mathcal{N}(\pm\mu_+^K, \sigma^2 I_d) \quad \text{if } y = +1, \qquad x \sim \mathcal{N}(\pm\mu_-^K, \sigma^2 I_d) \quad \text{if } y = -1,$$

with the following assumptions:

$$\mu_+^K \perp \mu_-^K, \quad \|\mu_+^K\| = \|\mu_-^K\| = \frac{1}{\sqrt{d}}, \quad \sigma = O\left(\frac{1}{\sqrt{d}\,\text{poly}\log(d)}\right).$$

Let

$$U := \frac{1}{n}\sum_{i=1}^n y_i(x_i^\top x_1)^2.$$

Fix $x_1, y_1$. For any $i \neq 1$, we write

$$x_1 = \mu_{y_1}^K + \varepsilon_1, \qquad x_i = \mu_{y_i}^K + \varepsilon_i,$$

with $\varepsilon_1, \varepsilon_i \sim \mathcal{N}(0, \sigma^2 I_d)$, independent. Then:

$$x_i^\top x_1 = {\mu_{y_i}^K}^\top \mu_{y_1}^K + {\mu_{y_i}^K}^\top \varepsilon_1 + {\mu_{y_1}^K}^\top \varepsilon_i + \varepsilon_i^\top \varepsilon_1.$$

Note that $({\mu_{y_i}^K}^\top \mu_{y_1}^K)^2 = \frac{1}{d^2}$ if $y_i = y_1$ and 0 otherwise.

Hence,

$$\mathbb{E}\left[y_i(x_i^\top x_1)^2 \mid y_i\right] = \begin{cases} \mathbb{E}[y_1(\pm\frac{1}{d} \pm {\mu_{y_1}^K}^\top \varepsilon_1 + {\mu_{y_1}^K}^\top \varepsilon_i + \varepsilon_i^\top \varepsilon_1)^2] & \text{if } y_i = y_1, \\ \mathbb{E}[-y_1(\pm{\mu_{-y_1}^K}^\top \varepsilon_1 + {\mu_{y_1}^K}^\top \varepsilon_i + \varepsilon_i^\top \varepsilon_1)^2] & \text{if } y_i \neq y_1. \end{cases}$$

Assuming a balanced distribution, i.e., $\Pr[y_i = y_1] = \Pr[y_i \neq y_1] = \frac{1}{2}$, we get:

$$\mathbb{E}\left[y_i(x_i^\top x_1)^2\right] = \frac{y_1}{2d^2} + O\left(\frac{1}{d^2 \cdot \text{poly}\log(d)}\right)$$

where in the above, we used ${\mu_{y_1}^K}^\top \varepsilon_1 = O(\frac{1}{d\cdot\text{poly}\log(d)})$ w.h.p. over the randomness in $\epsilon_1$.

Thus, the overall expectation is the following:

$$\mathbb{E}[U] = \frac{y_1}{2d^2} + O\left(\frac{1}{d^2 \cdot \text{poly}\log(d)}\right)$$

which aligns with the true label $y_1$.

To compute the finite sample guarantees, note that each summand

$$Z_i = y_i(x_i^\top x_1)^2$$

is sub-exponential with scale parameter $O(1/d)$ as $(\epsilon_i^\top \epsilon_1)^2$ has standard deviation $\frac{1}{d\text{poly}\log(d)}$ uniformly for all $i > 1$. By Bernstein's inequality, for any $\delta \in (0,1)$, with probability at least $1 - \delta$ over the randomness in $\{x_i, y_i\}_{i\in[n]}$,

$$|U - \mathbb{E}[U]| \leq C\left(\frac{1/d}{\sqrt{n}}\sqrt{\log(1/\delta)} + \frac{1/d}{n}\log(1/\delta)\right) = O\left(\frac{1}{d\sqrt{n}}\sqrt{\log(1/\delta)}\right),$$

for some absolute constant $C > 0$.

Putting together, with probability at least $1 - \delta$

$$U = \frac{1}{n}\sum_{i=1}^n y_i(x_i^\top x_1)^2 = \frac{y_1}{2d^2} \pm O\left(\frac{1}{d^2 \cdot \text{poly}\log(d)} + \frac{1}{d\sqrt{n}}\sqrt{\log(1/\delta)}\right).$$

In particular, if $n \gg d^2\log(1/\delta)$, then the error term is much smaller than the signal $\frac{1}{2d^2}$, and therefore $\text{sign}(T) = y_1$. with a union bound over all training points which introduces an additional factor $log(n)$ in the above bound, we find that the train error is exactly zero.

**Case II: $k \neq K$.** Now we evaluate the other terms in the summation in Eq. 14

$$x^\top A_j x = \frac{1}{n} \sum_{i=1}^{n} y_i^j (x^\top x_i^j)^2$$

for some $j \neq K$. drop 1 and note that

$$x_i \sim \mathcal{N}(\pm\mu_+^j, \ \sigma^2 I_d) \quad \text{if } y_i = +1, \quad x_i \sim \mathcal{N}(\pm\mu_-^j, \ \sigma^2 I_d) \quad \text{if } y_i = -1,$$
$$x \sim \mathcal{N}(\pm\mu_+^K, \ \sigma^2 I_d) \quad \text{if } y = +1, \quad x \sim \mathcal{N}(\pm\mu_-^K, \ \sigma^2 I_d) \quad \text{if } y = -1,$$

where $\mu_+^j, \mu_-^j, \mu_+^K, \mu_+^K$ are mutually orthogonal, $\quad \|\mu_+^j\| = \|\mu_-^j\| = \|\mu_+^K\| = \|\mu_-^K\| = \frac{1}{\sqrt{d}}$, and $\sigma = O\left(\frac{1}{\sqrt{d}\,\text{poly}\log(d)}\right)$.

Let

$$U' = \frac{1}{n} \sum_{i=1}^{n} y_i (x_i^\top x)^2.$$

let $x = \mu + \epsilon$, we have

$$\mathbb{E}\left[y_i (x_i^\top x)^2 \mid y_i\right] = \begin{cases} \mathbb{E}[(\pm\mu_{y_i}^{K}{}^\top \varepsilon + \mu^\top \varepsilon_i + \varepsilon_i^\top \varepsilon)^2] & \text{if } y_i = 1, \\ \mathbb{E}[-(\pm\mu_{-y_i}^{K}{}^\top \varepsilon + \mu^\top \varepsilon_i + \varepsilon_i^\top \varepsilon)^2] & \text{if } y_i = -1. \end{cases}$$

Hence,

$$\mathbb{E}[U'] = O\left(\frac{1}{d^2 \text{poly}\log(d)}\right).$$

Define

$$Z_i = y_i (x_i^\top x)^2.$$

By expanding $x_i = \mu_{y_i}^j + \varepsilon_i$, $x = \mu_y^K + \varepsilon'$, and using $\sigma = O(1/\sqrt{d})$, one can verify that

$$\text{Var}(x_i^\top x) = O\left(\frac{1}{d}\right),$$

and that $(x_i^\top x)^2$ is sub-exponential with scale parameter $O(1/d)$. Thus each $Z_i$ is sub-exponential with parameter $O(1/d)$. By Bernstein's inequality for i.i.d. sub-exponential random variables, for any $\delta \in (0,1)$, with probability at least $1 - \delta$,

$$|U'| = \left|\frac{1}{n} \sum_{i=1}^{n} (Z_i - \mathbb{E}[Z_i])\right| \leq C\left(\frac{1/d}{\sqrt{n}}\sqrt{\log(1/\delta)} + \frac{1/d}{n}\log(1/\delta)\right) = O\left(\frac{1}{d\sqrt{n}}\sqrt{\log(1/\delta)}\right),$$

for some absolute constant $C$.

**Combining the two cases.** Together with the two results above we find for any training data point $(x, y)$ from task $K$:

$$x^\top \left(\sum_{j=1}^{K} A_j\right) x = \frac{y}{2d^2} \pm O\left(\frac{\sqrt{K}}{d^2 \, \text{poly}\log(d)} + \frac{\sqrt{K}}{d\sqrt{n}}\sqrt{\log(1/\delta)}\right)$$

This concludes the calculations of the first term in Eq. 13.

Now let us consider the noise term (denoted by N) in Eq. 13:

$$N = \frac{1}{\sqrt{m}} \sum_{i=1}^{m} \omega_i (z_i^\top x)^2.$$

note that $\omega_i(z_i^\top x)^2$ has variance $O(\frac{1}{\text{poly}\log(d)})$, therefore by CLT

$$\frac{1}{\sqrt{m}}\sum_{i=1}^m \omega_i(z_i^\top x)^2 \to \mathcal{N}(0, \frac{1}{\text{poly}\log(d)}).$$

Overall, in the infinite width limit, for some $x, y$ from the $K$th task's empirical distribution

$$\Phi(w_{KT}, x) = \eta T\, x^\top(\sum_{j=1}^K A_j)x + \mathcal{N}(0,1)$$

$$= \eta T\left(\frac{y_k}{d^2} \pm O(\frac{\sqrt{K}}{d^2\,\text{poly}\log(d)} + \frac{\sqrt{K}}{d\sqrt{n}}\sqrt{\log(1/\delta))})\right) + O\left(\frac{\sqrt{\log(1/\delta)}}{\text{poly}\log(d)}\right).$$

In particular, if $n = \Omega(d^2 K\log(1/\delta))$, then the error term is smaller than the signal $\frac{y_k}{d^2}$, and if $\eta T = \Theta(d^2)$ then the output aligns with $y$. With a union bound over all training points (which introduces an additional factor $\log(n)$ in the above bound), we find that the train error (%) is exactly zero for all $k \in [n]$, leading to the zero train error.

## C.2. Characterizing Forgetting for Infinitely Wide Nets

We can directly compute $\widehat{F}_k(w_{KT})$ by computing $\Phi(w_{KT}, x_1^k)$ where $x_1^k$ is a sample (first sample w.l.o.g) from the training data for task $k$ where $k < K$. Recall,

$$w_{KT}^i \sim z + \frac{\eta T}{\sqrt{m}}\sum_{j=1}^K A_j\omega z, \quad A_j := \frac{1}{n}\sum_{v=1}^n y_v^j x_v^j {x_v^j}^\top$$

note that the above is symmetric with respect to the task index therefore $\lim_{m\to\infty}\Phi(w_{KT}, x_1^k) = \lim_{m\to\infty}\Phi(w_{KT}, x_1^K)$ in distribution. and we have in the $m \to \infty$ limit for $x_k := x_k^1$:

$$\Phi(w_{KT}, x_k) = \eta T\, x_k^\top(\sum_j A_j)x_k + \mathcal{N}(0, \frac{1}{\text{poly}\log(d)}) \tag{15}$$

$$= \eta T\left(\frac{y_k}{d^2} \pm O\left(\frac{\sqrt{K}}{d^2\,\text{poly}\log(d)} + \frac{\sqrt{K}}{d\sqrt{n}}\sqrt{\log(1/\delta)}\right)\right) + O\left(\frac{\sqrt{\log(1/\delta)}}{\text{poly}\log(d)}\right).$$

Therefore, again if $n = \Omega(d^2 K\log(1/\delta))$, then the error term is smaller than the signal $\frac{y_k}{d^2}$, and if $\eta T = \Theta(d^2)$ then the output aligns with $y_k$. With a union bound over all training points $k \in [n]$, we find that the training error is exactly zero for all tasks.

Now to characterize forgetting, recall it is defined as

$$|\widehat{F}_k(w_K) - \widehat{F}_k(w_k)| = |\sum_{x_k}\eta T\, x_k^\top(\sum_{j=k+1}^K A_j)x_k| \tag{16}$$

$$= \eta T \cdot O\left(\frac{\sqrt{K-k}}{d^2\,\text{poly}\log(d)} + \frac{\sqrt{K-k}}{d\sqrt{n}}\sqrt{\log(1/\delta)}\right).$$

where the calculations are the same as before except that the impact of initialization noise is present in both $\widehat{F}_k(w_K), \widehat{F}_k(w_k)$ and thus it is canceled.

In the above expression, if $n = \widetilde{\Omega}(d^2(K-k))$ and $\eta T \asymp d^2$, the increase in forgetting is $o_d(1)$.

## C.3. Finite-width Error

The calculations above hold for the infinitely-wide network. In this section, we derive the error due to finite width. Recall,

$$\Phi(w, x) = \Phi(w_0, x) + \frac{1}{\sqrt{m}}\sum_{i=1}^m a_i\phi'(\langle w_0^i, x\rangle)\langle x, w^i - w_0^i\rangle + O(\frac{\|w - w_0\|^2}{\sqrt{m}})$$

for the infinite width limit we had,

$$\bar{w}_1^i = w_0^i + \eta \frac{1}{\sqrt{m}} \frac{1}{n} \sum_{j=1}^n a_i \phi'(\langle w_0^i, x_j^1 \rangle) x_j^1 y_j^1$$

$$\bar{w}_t^i = w_0^i + \frac{\eta t}{\sqrt{m}} \frac{1}{n} \sum_{j=1}^n a_i \phi'(\langle w_0^i, x_j^1 \rangle) x_j^1 y_j^1$$

$$\bar{w}_{2t}^i = w_0^i + \frac{\eta t}{\sqrt{m}} \frac{1}{n} \sum_{j=1}^n a_i \phi'(\langle w_0^i, x_j^1 \rangle) x_j^1 y_j^1 + \frac{\eta t}{\sqrt{m}} \frac{1}{n} \sum_{j=1}^n a_i \phi'(\langle w_0^i, x_j^2 \rangle) x_j^2 y_j^2,$$

and similarly, all tasks' updates were derived. Let $\Phi(\cdot, \cdot)$ be the infinite-width and $\Phi_m(\cdot, \cdot)$ be the finite-width formulations of the network output. Then, we are interested in bounding $|\Phi(\bar{w}_t, x) - \Phi_m(w_t, x)|$ which can be written as:

$$|\Phi(\bar{w}_t, x) - \Phi_m(w_t, x)| \leq |\Phi(z, x) - \Phi_m(w_0, x)|$$

$$+ \left| \frac{1}{\sqrt{m}} \sum_{i=1}^m a_i \langle w_0^i, x \rangle \langle x, w_t^i - w_0^i \rangle - \eta t \mathbb{E}_z[\langle z, x \rangle \langle x, A_1 z \rangle] \right|$$

$$+ O(\frac{\|w_t - w_0\|^2}{\sqrt{m}})$$

$$\leq O(\frac{1}{\sqrt{m}} + \frac{\|w_t - w_0\|^2}{\sqrt{m}})$$

$$+ \left| \frac{1}{\sqrt{m}} \sum_{i=1}^m a_i \langle w_0^i, x \rangle \langle x, w_t^i - w_0^i \rangle - \eta t \mathbb{E}_z[\langle z, x \rangle \langle x, A_1 z \rangle] \right|$$

where we used the fact that by LLN: $\frac{1}{\sqrt{m}} \sum_{i=1}^m a_i \langle w_0^i, x \rangle \langle x, w_t^i - w_0^i \rangle \to \eta t \mathbb{E}_z[\langle z, x \rangle \langle x, A_1 z \rangle]$.

Note that $w_t^i - w_0^i = \frac{\eta}{\sqrt{mn}} \sum_{\tau=0}^{t-1} \sum_{j=1}^n a_i \langle w_\tau^i, x_j \rangle x_j y_j$, therefore when $m \to \infty$:

$$\frac{1}{\sqrt{m}} \sum_{i=1}^m a_i \langle w_0^i, x \rangle \langle x, w_t^i - w_0^i \rangle = \frac{\eta}{n} \sum_{\tau=0}^{t-1} \sum_{j=1}^n \langle x, x_j y_j \rangle \frac{1}{m} \sum_{i=1}^m \langle w_0^i, x \rangle \langle w_\tau^i, x_j \rangle$$

$$\to \frac{\eta}{n} \sum_{\tau=0}^{t-1} \sum_{j=1}^n \langle x, x_j y_j \rangle \mathbb{E}[\langle w_0^i, x \rangle \langle w_\tau^i, x_j \rangle]$$

$\langle w_0^i.x \rangle$ is Gaussian with variance $\|x\|^2$ and $\langle w_\tau^i, x_j \rangle$ is bounded by $D_\tau^i \|x_j\|$ where $D_\tau^i := \|w_\tau^i - w_0^i\|$, therefore $\langle w_0^i, x \rangle \langle w_\tau^i, x_j \rangle$ is bounded by $\|x\| \|x_j\| D_\tau^i = O(D_\tau^i)$. and by Hoeffding's concentration inequality:

$$\left| \frac{1}{m} \sum_{i=1}^m \langle w_0^i, x \rangle \langle w_\tau^i, x_j \rangle - \mathbb{E}[\langle w_0^i, x \rangle \langle w_\tau^i, x_j \rangle] \right| = O(\frac{D_\tau^i}{\sqrt{m}})$$

and hence w.h.p,

$$\left| \frac{\eta}{n} \sum_{\tau=0}^{t-1} \sum_{j=1}^n \langle x, x_j y_j \rangle \frac{1}{m} \sum_{i=1}^m \langle w_0^i, x \rangle \langle w_\tau^i, x_j \rangle - \frac{\eta}{n} \sum_{\tau=0}^{t-1} \sum_{j=1}^n \langle x, x_j y_j \rangle \mathbb{E}[\langle w_0^i, x \rangle \langle w_\tau^i, x_j \rangle] \right|$$

$$= O(\frac{\eta}{\sqrt{m}} \sum_\tau \max_i D_\tau^i)$$

$$= \tilde{O}(\frac{\eta}{\sqrt{m}} \sum_\tau D_\tau^1) = \tilde{O}(\frac{\eta t D_t^1}{\sqrt{m}})$$

where we used again $x^\top x_j \lesssim 1$, the fact that due to symmetry we expect $D_\tau^i$ to be of the same order for different $i$ s and also $D_\tau^i < D_t^i$ for all $\tau \leq t$. Putting these back to the inequality in the last page for the finite-width error of the network's output:

$$|\Phi(\bar{w}_t, x) - \Phi_m(w_t, x)| = \tilde{O}\left( \frac{1}{\sqrt{m}} + \frac{\|w_t - w_0\|^2}{\sqrt{m}} + \frac{\eta t \|w_t^1 - w_0^1\|}{\sqrt{m}} \right).$$

similarly

$$|\Phi(\bar{w}_{KT}, x) - \Phi_m(w_{KT}, x)| = \tilde{O}\left(\frac{1}{\sqrt{m}} + \frac{\|w_{KT} - w_0\|^2}{\sqrt{m}} + \frac{\eta KT\|w_{KT}^1 - w_0^1\|}{\sqrt{m}}\right). \tag{17}$$

### C.3.1. BOUNDING THE WEIGHTS DISTANCE FROM INITIALIZATION

In order to complete the proof, we need to bound the distance from initialization i.e., $\|w_t - w_0\|$ and $\|w_t^i - w_0^i\|$ for every $i$ and $t$. We do this by an iterative argument as follows. Note that for the XOR cluster dataset $\|x\| = \Theta_d(1)$. Then, by recalling the updates of GD, we find that,

$$\|w_1^i - w_0^i\| \le \frac{\eta}{\sqrt{mn}} \sum_{i=1}^{n} |\langle w_0^i, x_j^1\rangle| \|x_j^1\| \lesssim \frac{\eta}{\sqrt{m}}$$

$$\|w_2^i - w_0^i\| \le \frac{\eta}{\sqrt{mn}} \sum_{i=1}^{n} |\langle w_0^i, x_j^1\rangle| \|x_j^1\| + \frac{\eta}{\sqrt{mn}} \sum_{i=1}^{n} |\langle w_1^i, x_j^1\rangle| \|x_j^1\|$$

$$\le \frac{2\eta}{\sqrt{mn}} \sum_{i=1}^{n} |\langle w_0^i, x_j^1\rangle| \|x_j^1\| + \frac{\eta}{\sqrt{mn}} \sum_{i=1}^{n} |\langle w_1^i - w_0^i, x_j^1\rangle| \|x_j^1\|$$

$$\lesssim \frac{2\eta}{\sqrt{m}} + \frac{\eta^2}{m} = O(\frac{2\eta}{\sqrt{m}})$$

$$\|w_3^i - w_0^i\| \le \frac{3\eta}{\sqrt{mn}} \sum_{i=1}^{n} |\langle w_0^i, x_j^1\rangle| \|x_j^1\| + \frac{\eta}{\sqrt{mn}} \sum_{i=1}^{n} |\langle w_1^i - w_0^i, x_j^1\rangle| \|x_j^1\|$$

$$+ \frac{\eta}{\sqrt{mn}} \sum_{i=1}^{n} |\langle w_2^i - w_0^i, x_j^1\rangle| \|x_j^1\|$$

$$\lesssim \frac{3\eta}{\sqrt{m}} + (\frac{\eta}{\sqrt{m}})^2 + (\frac{\eta}{\sqrt{m}})^3 = O(\frac{3\eta}{\sqrt{m}})$$

Therefore, $\|w_t^i - w_0^i\| = O(\frac{t\eta}{\sqrt{m}})$ when $\eta = O_m(1)$. We also have

$$\|w_t - w_0\| = O(t\eta).$$

By Eq. 17:

$$|\Phi(\bar{w}_{KT}, x) - \Phi_m(w_{KT}, x)| = \tilde{O}\left(\frac{1}{\sqrt{m}} + \frac{\eta^2 K^2 T^2}{\sqrt{m}} + \frac{\eta^2 K^2 T^2}{m}\right) = \tilde{O}(\frac{\eta^2 K^2 T^2}{\sqrt{m}}) \tag{18}$$

Recall that we had chosen $\eta T = \Theta(d^2)$ to guarantee $\text{sign}(\Phi(\bar{w}_{KT}, x_k)) = y_k$ and $|\Phi(\bar{w}_{KT}, x_k)| \gtrsim 1$, therefore if

$$m = \tilde{\Omega}(d^8 K^4),$$

the finite width error is small enough to conclude $\text{sign}(\Phi_m(w_{KT}, x_K)) = y_K$ for any $x_K, y_K$ from the $K$th task data distribution. Similarly, we have $\text{sign}(\Phi_m(w_{KT}, x_k)) = y_k$ for any $x_k, y_k$ from the $k$th task's data distribution because the error terms defined above are independent of the data distribution. Thus, the characterization of forgetting we derived in Eq. 15 is accurate for the same width.

Finally, we note that with the given assumptions on $n, T, m, K$ it holds that $\Phi_m(w_{KT}, x)$ is always bounded by 1. To see this, recall by Eq. 15 and Eq. 18, the network output for any training point $x$ is at most hte following:

$$\Phi_m(w_{KT}, x) \le \eta T\left(\frac{y_k}{d^2} \pm O\left(\frac{\sqrt{K}}{d^2 \text{ poly} \log(d)} + \frac{\sqrt{K}}{d\sqrt{n}}\sqrt{\log(1/\delta)}\right)\right) + O\left(\frac{\sqrt{\log(1/\delta)}}{\text{poly} \log(d)}\right)$$

$$+ \tilde{O}\left(\frac{\eta^2 K^2 T^2}{\sqrt{m}}\right)$$

Recall $K = \tilde{O}_d(1)$, with the choice of $m, n$ in the statement of the theorems, it holds with high probability that $\Phi_m(w_{KT}, x) \le 1$. Thus, the network output always lies in the linear part of the hinge-loss for any $\eta T \le d^2$ even at initialization where $T = 0$. Therefore, our assumption on the linearity of loss is valid throughout training.

# D. Regularized Continual Learning: Proof of Proposition 2.7

**Proposition D.1** (Restatement of Prop. 2.7). *Consider the regularized continual learning problem Eq.5 with same setup as Theorem 2.1 with $m \to \infty$. The iterates of this algorithm with step-size $\eta$ are equivalent to unregularized continual learning with step-size $\widetilde{\eta}_T$ where $\widetilde{\eta}_T = \alpha_T \eta / T$ and $\alpha_T = \frac{1 - (1 - \eta\lambda)^T}{\eta\lambda}$.*

*Proof.* In regularized continual learning, the objective at task $k \geq 2$ is:

$$\min_w \widehat{F}_k(w) + \frac{\lambda}{2} \|w - w_{k-1}\|^2$$

The GD update rule is the following:

$$
\begin{aligned}
w_k^{(t+1)} &= w_k^{(t)} - \eta \nabla \widehat{F}_k(w_k^{(t)}) - \eta\lambda(w_k^{(t)} - w_{k-1}) \\
&= (1 - \eta\lambda)w_k^{(t)} - \eta \nabla \widehat{F}_k(w_k^t) + \eta\lambda w_{k-1}.
\end{aligned}
$$

For the first task, there is no regularization, therefore for neuron $i$ (we drop $i$ here for ease of notation):

$$w_1^{(1)} = w_1^{(0)} + \eta \frac{1}{\sqrt{m}} \frac{1}{n} \sum_{j=1}^n a_i \phi'(\langle w_1^{(0)}, x_j^1 \rangle) x_j^1 y_j^1$$

$$w_1 := w_2^{(0)} = w_1^{(T)} = w_1^{(0)} + \frac{\eta T}{\sqrt{m}} \frac{1}{n} \sum_{j=1}^n a_i \phi'(\langle w_1^{(0)}, x_j^1 \rangle) x_j^1 y_j^1$$

For the second task, due to the regularization term $\lambda \|w - w_1\|^2 / 2$, the first GD update takes the following shape:

$$
\begin{aligned}
w_2^{(1)} &= (1 - \eta\lambda)w_2^{(0)} + \eta \frac{1}{\sqrt{m}} \frac{1}{n} \sum_{j=1}^n a_i \phi'(\langle w_1^{(0)}, x_j^2 \rangle) x_j^2 y_j^2 + \eta\lambda w_1 \\
&= w_2^{(0)} + \eta \frac{1}{\sqrt{m}} \frac{1}{n} \sum_{j=1}^n a_i \phi'(\langle w_1^{(0)}, x_j^2 \rangle) x_j^2 y_j^2.
\end{aligned}
$$

Hence, the first step is identical to the unregularized update rule. For the second step,

$$
\begin{aligned}
w_2^{(2)} &= (1 - \eta\lambda)w_2^{(1)} + \eta \frac{1}{\sqrt{m}} \frac{1}{n} \sum_{j=1}^n a_i \phi'(\langle w_1^{(0)}, x_j^2 \rangle) x_j^2 y_j^2 + \eta\lambda w_1 \\
&= w_2^{(0)} + ((1 - \eta\lambda) + 1) \frac{\eta}{\sqrt{m}} \frac{1}{n} \sum_{j=1}^n a_i \phi'(\langle w_1^{(0)}, x_j^2 \rangle) x_j^2 y_j^2.
\end{aligned}
$$

Similarly,

$$
\begin{aligned}
w_2^{(3)} &= (1 - \eta\lambda)w_2^{(2)} + \eta \frac{1}{\sqrt{m}} \frac{1}{n} \sum_{j=1}^n a_i \phi'(\langle w_1^{(0)}, x_j^2 \rangle) x_j^2 y_j^2 + \eta\lambda w_1 \\
&= w_2^{(0)} + ((1 - \eta\lambda)((1 - \eta\lambda) + 1) + 1) \frac{\eta}{\sqrt{m}} \frac{1}{n} \sum_{j=1}^n a_i \phi'(\langle w_1^{(0)}, x_j^2 \rangle) x_j^2 y_j^2.
\end{aligned}
$$

Therefore for $t \leq T$:

$$w_2^{(t)} = w_2^{(0)} + \alpha_t \frac{\eta}{\sqrt{m}} \frac{1}{n} \sum_{j=1}^n a_i \phi'(\langle w_1^{(0)}, x_j^2 \rangle) x_j^2 y_j^2.$$

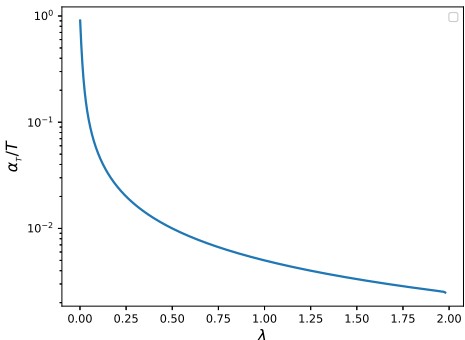 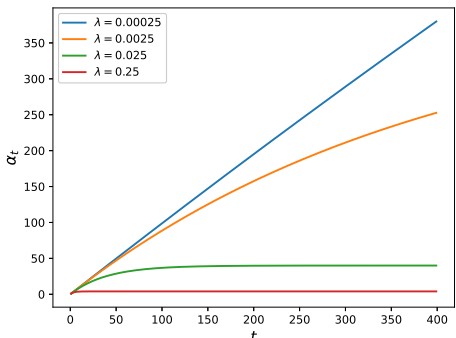

*Figure 8.* Effective step-size for regularized continual learning $\alpha_t$ in Prop. 2.7 based on regularization parameter $\lambda$ (Left) and number of GD steps $t$ (Right).

The same steps can be repeated for every task to obtain:

$$w_k^{(t)} = w_k^{(0)} + \alpha_t \frac{\eta}{\sqrt{m}} \frac{1}{n} \sum_{j=1}^{n} a_i \phi'(\langle w_1^{(0)}, x_j^2 \rangle) x_j^2 y_j^2,$$

which leads to the following expression for any $k \geq 2$ :

$$w_k := w_1^{(0)} + \frac{\eta T}{\sqrt{m}} \frac{1}{n} \sum_{j=1}^{n} a_i \phi'(\langle w_1^{(0)}, x_j^1 \rangle) x_j^1 y_j^1 + \alpha_T \frac{\eta}{\sqrt{m}} \frac{1}{n} \sum_{\kappa=2}^{k} \sum_{j=1}^{n} a_i \phi'(\langle w_1^{(0)}, x_j^\kappa \rangle) x_j^\kappa y_j^\kappa$$

where

$$\alpha_1 = 1, \alpha_t = (1 - \eta\lambda)\alpha_{t-1} + 1 \ \ \text{for} \ \ t > 1$$

We can find the following closed form expression to the equations above: $\alpha_t = \frac{1-(1-\eta\lambda)^t}{\eta\lambda}$. This completes the proof. $\quad\square$

With an accurate approximation, we have

$$\alpha_t \approx \frac{1 - e^{-\eta\lambda t}}{\eta\lambda}.$$

For small $t$, we have $\alpha_t \approx t$, whereas for large $t \approx T$, assuming $\lambda = c/T$ : we have $\alpha_t = \frac{T(1-e^{-\eta c})}{\eta c}$. Fig. 8 illustrates $\alpha_T/T$ versus regularization parameter $\lambda$ and $\alpha_t$ based on $t$ for different regularization parameters. Note that larger values of $\lambda$ correspond to smaller values of $\alpha_t$ leading to weights moving shorter distances from their initialization points. As $\lambda \to 0$, we have $\alpha_T/T \to 1$, as the step-size for regularized problem converges to the step-size for unregularized one.

## E. Additional Experiments and Implementation Details

**Experiments with transformers and GMM data.** We also conduct experiments on attention-based architecture in Fig. 12. We plot the train-time forgetting for task 1 for $K = 2$ overall tasks for a transformer with feedforward neural networks in both the encoder and the decoder parts where we consider $m_{encoder} = 60, m_{decoder} = 30$ for the left plot and $m_{encoder} = m_{decoder} = 10$ for the right plot. Results shown are averaged over 10 independent experiments. We remark that for the transformer with smaller size, we observe the similar behavior we observed for neural network experiments, i.e, increasing the sample-size for the second task can noticeably help with train-time forgetting of the first task. On the other hand, for the larger network, the behavior is more complex: increasing $n$ can help up to a certain threshold ($n \approx 250$), while above this threshold increasing $n$ hurts continual learning. While we hypothesize this behavior is due to the complex landscape of larger networks, a more thorough investigation is needed.

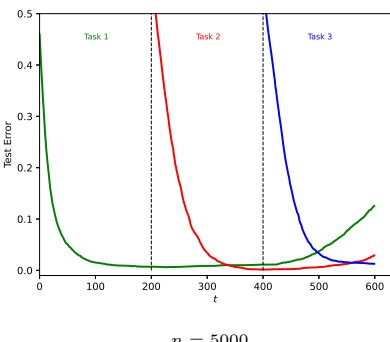 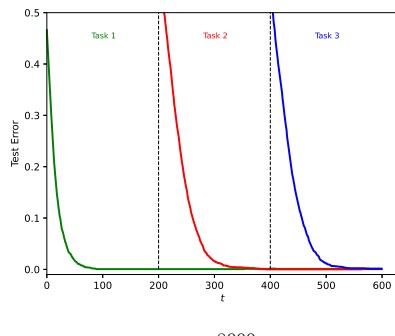

$$n = 5000 \qquad\qquad n = 8000$$

*Figure 9.* Classification test error for each task vs iterations for the XOR cluster with $K = 3$ tasks trained on a quadratic network with $n = 5000$(left) and $n = 8000$(right) training samples per task.

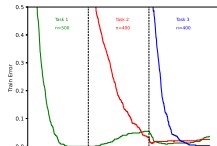 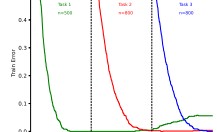 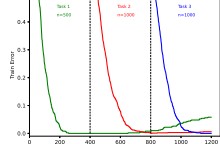 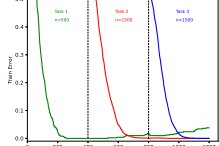 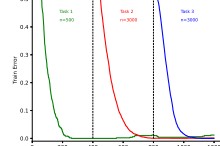

*Figure 10.* Repeating the experiment of Fig. 3 but with ReLU activation and logistic loss.

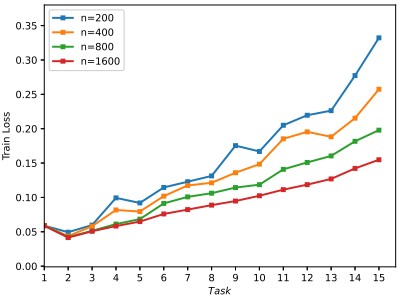 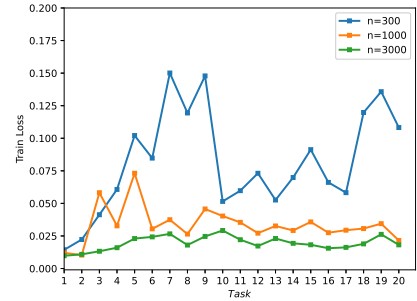

*Figure 11.* Train loss on task 1 as a function of the task index (i.e., $\widehat{F}_1(w_k)$ vs. $k$) for $K = 15$ and $K = 20$ tasks with $n$ samples per task for the XOR cluster dataset. The left plot uses GELU activation with logistic loss, while the right plot uses quadratic activation with hinge loss.

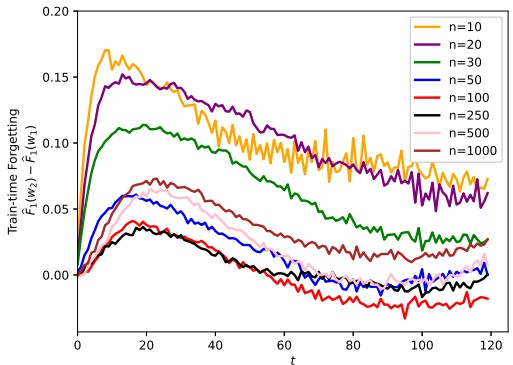 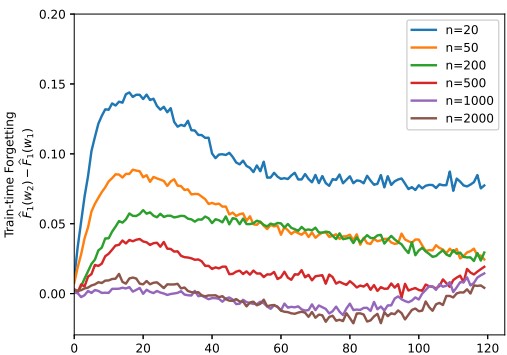

*Figure 12.* Train-time forgetting for task 1 based on $t$ for $K = 2$ tasks with attention-based transformers with large neural net for the encoder and decoder parts(Left plot), and with small neural net(Right plot). Here we consider a tokenized multi-task Gaussian-mixture data where the goal is to find the binary label used for each context window. We fix $n = 50$ for the first task and change $n$ for the second task. Note that our insights from previous theoretical and empirical results partially hold for this setting, especially for the transformer with smaller FFN layer.

**Implementation Details for all experiments.** We include the actual values for different problem parameters used in the numerical experiments:

Fig. 1: $n = 2500$ (left), $n = 5000$(right), for both plots we set $d = 50, m = 1000, \eta = 2, T = 200, \sigma = 0.1/\sqrt{d}$ and use linear loss and quadratic activation.

Fig. 2: $d = 50, m = 1000, \eta = 2, T = 200, \sigma = 0.1/\sqrt{d}$.

Fig. 3: GELU activation and logistic loss. $d = 50, m = 400, \eta = 3, T = 400, \sigma = 0.1/\sqrt{d}$.

Fig. 4: GELU activation, logisitc loss for both plots. We set $d = 50, m = 2000, \eta = 30, T = 2000, \sigma = 0.2/\sqrt{d}$. Right: $n = 2000, T = 4000$.

Fig. 6: We set $n = 5000, d = 75, \eta = 5, T = 200, \sigma = 0.15/\sqrt{d}$ and vary $m = 100, 300, 1000, 3000, 6000, 10000$.

Fig. 5: Top: $n = 50$ samples for the first task, $n$ varying for the second task, GELU activation, Hinge loss, $d = 784, m = 500$. For the left plot $\eta = 0.0003, T = 50$ and for the right $\eta = 0.001, T = 200$. The results are averages over 15 experiments. Bottom: $T = 2000, \eta = 0.05, m = 2000, K = 4$, ReLU activation and Logistic loss, Tasks are chosen from labels 1-4, 7-10 from the FMNIST dataset. Dataset is normalized to have $\ell_2$-norm at most 1.

Fig. 7: GELU activation, Logistic loss, $d = 50, n = 200, \eta = 20, T = 1000, \sigma = 0.2/\sqrt{d}$

Fig. 9: $n = 5000$(left),$8000$(right),$d = 75, m = 1000, \eta = 5, T = 200, \sigma = 0.15/\sqrt{d}$, linear loss, quadratic activation

Fig.10: Using the same setup as Fig. 3 but with ReLU activation and logistic loss. $d = 50, m = 1000, \eta = 0.3, T = 400, \sigma = 0.1/\sqrt{d}$

Fig. 11: GELU activation and logisitc loss, $\eta = 30, m = 400$ for the left plot, Quadratic activation and Hinge loss, $m = 1000, \eta = 4$ for the right plot. For both plots we set, $d = 50, T = 400, \sigma = 0.1/\sqrt{d}$.

Fig. 12: We use hinge-loss, ReLU activation, and the transformer is one-layer with one head, context length = 10, the hidden-layer size of the feedforward neural is 60 and for the decoder is 30. In the right plot, both hidden-layer sizes are reduced to 10 $\sigma = 0.1/\sqrt{d}, \mu^k = \mathbf{e}_k/\sqrt{d}$ for $k \in [2], \eta = 0.01$.

