# OpenReview forum: "On the Theory of Continual Learning with Gradient Descent for Neural Networks"
_ICML.cc/2026/Conference — ICML 2026 regular_

### Official Review · Reviewer_jE2c · 2026-03-12

**Soundness:** 2
**Presentation:** 3
**Significance:** 2
**Originality:** 2
**Overall Recommendation:** 2
**Confidence:** 4

**Summary:**

This paper studies catastrophic forgetting in neural networks trained sequentially with gradient descent. The authors analyze a stylized setting consisting of a two-layer neural network with quadratic activation trained on a sequence of XOR-cluster classification tasks. They derive closed-form bounds for both train-time and test-time forgetting as functions of key parameters such as network width, number of tasks, sample size per task, and training horizon. The analysis decomposes test-time forgetting into a train-time forgetting term and a delayed generalization gap and bounds these terms using a combination of gradient descent dynamics and algorithmic stability arguments. Numerical experiments on synthetic datasets and small neural networks qualitatively support the theoretical predictions.

A central concept assessed by the manuscript is the theoretical characterization of catastrophic forgetting in neural networks trained with gradient descent. Overall, a central domain examined by this manuscript is the theoretical understanding of continual learning beyond linear models.

**Compliance With Llm Reviewing Policy:**

Affirmed.

**Key Questions For Authors:**

It would help clarify what the authors view as the main conceptual insight beyond deriving explicit bounds. Do the results reveal a mechanism of forgetting that was previously unknown, or do they primarily formalize existing empirical observations? Additionally, how sensitive are the conclusions to the XOR-cluster data model and the kernel regime assumptions?

**Limitations:**

The theory is derived in a very restricted setting (two-layer networks, NTK regime, orthogonal tasks), and it is unclear how well the conclusions extend to modern deep networks operating in the feature-learning regime.

**Strengths And Weaknesses:**

Strengths. The paper tackles an important theoretical question: understanding the mechanisms of catastrophic forgetting in neural networks. The analysis connects several important quantities—sample size, network width, number of tasks, and training horizon—and derives explicit bounds relating them to forgetting. The decomposition of forgetting into optimization and generalization components is conceptually useful. The theoretical predictions are also qualitatively supported by experiments.

Weaknesses. The main conceptual advance of the paper is somewhat unclear. While the authors derive explicit bounds, the qualitative conclusions largely align with existing intuition: larger datasets, wider networks, and early stopping can mitigate forgetting. The analysis also relies on a highly stylized setting (two-layer networks with quadratic activation in the NTK regime and orthogonal XOR tasks), making it difficult to assess how much of the theory carries over to realistic deep learning systems.

---

> ### Author Rebuttal · Authors · 2026-03-31
>
> We thank the reviewer for the careful reading, for their insightful comments & questions, and for highlighting the contributions of our work. We address the weaknesses, limitations and questions below and we would be happy to clarify any further points.
>
>
> Regarding the **conceptual message** of the paper, our view is that the main conceptual insight is not only that larger width, more samples, and early stopping help. Rather, the paper identifies a characterization of forgetting for neural nets in the kernel regime and identifies a positive regime (with orthogonal tasks) and a mechanism for successful continual learning with plain sequential GD in a nonlinear neural network. Concretely, we show that in the kernel regime, unregularized sequential GD can achieve arbitrarily small forgetting and small test error for all tasks simultaneously under explicit scaling conditions and furthermore L-2 regularization cannot help improve the derived scalings in the studied setting.
>
> We agree that extending the results beyond the kernel regime is an important next step. At the same time, even this kernel regime yields several nontrivial conclusions: (1) continual learning can be possible without explicit regularization, particularly when tasks are independent and (near-) orthogonal (2) forgetting admits a clean decomposition into optimization and delayed-generalization components, and (3) later-task sample size can improve retention of earlier tasks. We therefore view the paper as a foundational characterization of continual learning beyond linear models, rather than as a complete theory of modern feature-learning regimes, especially given that the theoretical results for continual learning with NNs in the literature are very limited. To the best of our knowledge, these findings constitute new contributions to the continual learning theory literature. We also note that kernel regime analyses can still be informative for some LLM fine-tuning settings, particularly parameter-efficient regimes in which the model stays close to the pretrained parameters. Thank you for pointing this out. We will include this discussion in the paper.
>
>
>
> Regarding the **limitations** of our setup: Extensions to the non-orthogonal tasks and other data distributions is possible. The choice of XOR data was motivated by the study of single/multi index models in the deep learning theory literature (cf. the discussion in section 2.1.2 in the paper) and our theorems are specific to a one-hidden-layer network, chosen because it is expressive enough to learn the XOR cluster data distribution while still permitting explicit control of the sequential GD dynamics. Specializing to a class of data also helps better capture the role of problem parameters such as number of samples. However, we expect similar insights on the role of problem parameters can be applicable to other multi-index models.  In fact the delayed-generalization bound in our analysis holds for almost any data distribution and our approach for the train-time forgetting shows that for general data, the train-time forgetting is characterized by (cf. the discussion after Thm. 2.2. and  also Eqs. (16)-(17) in the appendix):
>
> $$\left|\widehat{F}\_k\left(w\_K\right)-\widehat{F}\_k\left(w\_k\right)\right|= |\sum_{x\_k} \eta T x\_k^{\top}(\sum_{j=k+1}^K A\_j) x\_k| + O(\frac{||w\_{K T}-w\_0||^2}{\sqrt{m}}),$$
>
> where we denote $A_j:=\frac{1}{n^2} \sum_{v=1}^n y_v^j x_v^j x_v^{j^{\top}}$ and $(x_v^j,y_v^j)_{v\in[n]} $ denotes the training data for task $j$. Our results in the theorems are obtained by specializing this bound to the xor cluster data distribution with Gaussian noise. However, the approach is applicable to various data distributions. We will include this as a remark in the next version of the paper. We thank the reviewer for highlighting this point.
>
>
> Finally, we find it noteworthy that although the results are mainly developed in the stylized setting, the experimental section already shows the predictions across different loss functions, activations, data distributions, architectures, step sizes, and training horizons. We will highlight this more clearly so the empirical section better communicates that the qualitative insights extend beyond the exact theorem assumptions.

---

### Official Review · Reviewer_LtxF · 2026-03-13

**Soundness:** 3
**Presentation:** 3
**Significance:** 2
**Originality:** 3
**Overall Recommendation:** 4
**Confidence:** 3

**Summary:**

This paper provides closed-form theoretical guarantees for forgetting in continual learning under a one-hidden-layer quadratic neural network and an XOR-cluster data setting. It analyzes train-time forgetting and test-time forgetting separately, and leverages algorithmic stability to bound the delayed generalization gap. The study reveals that the interaction between the number of samples, network width, and early stopping determines the degree of forgetting, and derives a parameter scaling regime that leads to vanishing forgetting.

**Compliance With Llm Reviewing Policy:**

Affirmed.

**Key Questions For Authors:**

1. Is the condition ( $m = \theta(d^8 K^4)$ ) tight? Is there any lower bound suggesting that the polynomial dependence on ( $K$ ) is necessary?
2. Could the implications of Proposition 2.6—showing that regularization methods (e.g., EWC) are ineffective in the kernel regime—be discussed more clearly?
3. What qualitative changes occur in the forgetting bound if the orthogonality assumption between tasks is relaxed?
4. Are there any conjectures regarding how network depth affects forgetting dynamics?

**Limitations:**

The extremely restricted model and data setting, the unrealistic width condition, and the limitations of the kernel regime are acknowledged, but a more explicit discussion of these issues is needed.

**Strengths And Weaknesses:**

# Strength
- Provides the first closed-form bound for forgetting in continual learning within neural networks
- The decomposition framework of test-time forgetting = train-time forgetting + delayed generalization gap is clear and elegant
- The finding that increasing width alone cannot eliminate forgetting contrasts with prior empirical claims and represents an important theoretical contribution
- Presents the counterintuitive result that increasing the number of samples in later tasks can reduce forgetting of earlier tasks

# Weakness
- The analysis relies on a highly restricted setting (one-hidden-layer quadratic network with XOR-cluster data), creating a large gap from real deep Transformers
- The width condition for vanishing forgetting, m = Θ(d⁸K⁴), is unrealistically large; it is unclear whether this reflects a limitation of the proof technique or a fundamental barrier
- The kernel/lazy regime analysis may fail to capture the feature learning dynamics that are important in practical continual learning
- The experiments are limited to the theoretical setting and lack validation on standard continual learning benchmarks (e.g., Split-CIFAR)

---

> ### Author Rebuttal · Authors · 2026-03-31
>
> We thank the reviewer for the careful reading, for their insightful comments & questions, and for highlighting the contributions of our work. We address the weaknesses and questions below and we would be happy to clarify any further points.
>
> **Q1 & W2** -  We agree that the sufficient scaling for width in our theorems is  large. Our main theorem gives a guarantee for vanishing forgetting in a nonlinear neural network in the kernel regime. At present, we do not have a matching lower bound showing that this polynomial dependence on $(d)$ and $(K)$ is unavoidable. Our view is that the large exponents mainly reflect the theoretical obstacles of obtaining a fully explicit closed-form analysis while keeping the dynamics in the kernel regime. We will clarify this point in the revision. We note that the empirical trends predicted by the theory already appear at substantially smaller widths in our experiments. We also note that previous works for standard stationary setups have also shown such large dependencies on the network width (e.g. see [Ji and Telgarsky 2020, Taheri and Thrampoulidis 2024]).
>
> **Q2** - Thank you for raising this point. Our exact claim is *not* that regularization-based continual-learning methods are ineffective in general. Rather, Proposition 1 shows a narrower statement: in the kernel regime considered in the paper, the $L_2$ penalty around the previous-task solution is equivalent to unregularized continual learning with a different effective step size. Therefore, within this regime, such regularization does not improve the sample scaling established by our theory. We will revise the discussion to make this scope more clear. At the same time, this result provides an intuitive conceptual message: in the lazy regime, successful continual learning can already occur without explicit regularization under suitable scaling (Thms.  2.1-2.3), because the iterates remain close to initialization and the dynamics are implicitly stabilized by the kernel regime itself.
>
> **Q3** - The orthogonality assumption is mainly used to eliminate cross-task interference terms and obtain a clean closed-form dependence on width, sample size, number of tasks, and training horizon. If orthogonality is relaxed, additional correlation terms involving overlaps between task means( $(\mu_+^k, \mu_+^j)$) enter optimization dynamics and bounds. We expect the same proof strategy to extend, but the rates can deteriorate with the degree of task overlap and the resulting expressions would be substantially more cumbersome. Please also see our response to **W3 & L2** from *Reviewer BBCJ*. We will make this more explicit in the revision.
>
> **Q4 & W1** - Our theorems are specific to a one-hidden-layer quadratic network, chosen because it is expressive enough to learn the XOR cluster data distribution while still permitting explicit control of the sequential GD dynamics. A natural conjecture is that additional depth could change the forgetting picture by modifying the effective features map by the network and potentially allowing less width for each layer. However, establishing this rigorously requires going beyond the current proof techniques. We will add a short discussion stating this future direction.
>
> **W3** - Extensions to the feature learning regime is highly desirable as it shows the benefits and downfalls of each regime for continual learning. We would like to note that even in this restricted setting, the paper establishes several nontrivial facts: test-time forgetting can be decomposed into train-time forgetting and delayed generalization gap; vanishing forgetting is possible in a nonlinear neural network without explicit regularization, particularly with orthogonal tasks; and later-task sample size can reduce forgetting of earlier tasks. We see an extension to feature-learning regimes as an important and complementary next direction, and we will highlight this more explicitly in the revised manuscript. Thank you for mentioning this point.
>
> **W4** - We agree that broader empirical validation on standard continual-learning benchmarks would strengthen the paper. Our goal in the current submission was not to propose a new benchmark-leading algorithm, but to validate the qualitative predictions of the theory in controlled settings where the roles of width, sample size, number of tasks, learning rate etc can be isolated clearly for realizable multi-index models in the studied kernel regime setup.. For this reason, we focused mainly on experiments close to the theoretical setup, while also including preliminary studies beyond the analyzed setting for different architectures, activations, loss functions, data distributions etc.

---

### Official Review · Reviewer_BBCJ · 2026-03-13

**Soundness:** 3
**Presentation:** 4
**Significance:** 4
**Originality:** 3
**Overall Recommendation:** 5
**Confidence:** 4

**Summary:**

The authors provide the theoretical characterization of catastrophic forgetting in neural networks trained via GD, specifically deriving closed-form bounds that link forgetting rates to sample size, network width, and training iterations.
Overall, a central domain examined by this manuscript is the intersection of optimization dynamics in the kernel regime and continual learning theory, moving beyond linear models to provide the first rigorous guarantees for non-linear neural networks on non-trivial data distributions. The authors analyze a series of orthogonal tasks and decompose test-time forgetting into training-time forgetting and delayed generalization gap.  They demonstrated that while overparameterization and large sample sizes can alleviate forgetting, neither alone is sufficient to eliminate forgetting.
Conversely, to achieve complete forgetting, a specific scaling relationship is required between network width, sample size, and number of iterations. The paper concludes that standard regularization techniques offer no additional benefit in this regime as they are equivalent to step-size rescaling.

**Compliance With Llm Reviewing Policy:**

Affirmed.

**Final Justification:**

My concerns have been adequately addressed. I keep the positive scroe.

**Key Questions For Authors:**

see weakness and limitations.

**Limitations:**

1. This analysis is strictly limited to inertial/kernel mechanisms where features remain largely invariant. However, empirical evidence suggests that successful sustained learning in large models often relies on feature adaptation (plasticity), not just linear readout stability. Because this theory is limited to feature-invariant mechanisms, it may overlook the core mechanisms of sustained learning in biological systems or large linear learning models (LLMs).

2. The theoretical bounds depend on the orthogonality of the task means (mu_k^+ is perpendicular to mu_j^+). If there is correlation between tasks (common in real life, e.g., learning "cat" before learning "dog"), the interference terms ignored in the proof may dominate, potentially invalidating the derived scaling laws. The paper acknowledges this but does not provide a bound for the non-orthogonal case.

3. The polynomial dependence on the dimension d and the number of tasks K (e.g., m is proportional to K^4) implies that the required width becomes extremely large even for a sequence containing only 10 tasks in a moderate dimension. This limits the theory's utility as a practical system design guide, making it more suitable as a proof of existence.

**Strengths And Weaknesses:**

Strengths
1. This work provides the first closed-form forgetting guarantees for two-layer quadratic neural networks trained by GD. Unlike prior work restricted to linear models or empirical observations, this paper derives explicit bounds showing how forgetting scales.

2. The authors proved and verified that the diminishing returns of model width and sample size must be adjusted in sync with the number of tasks and training cycles. The finding that increasing the sample size of subsequent tasks can reduce forgetting of previous tasks is counterintuitive but highly insightful.

3.  The analytical framework decomposes test-time forgetting into training-time forgetting  (optimization error) and delayed generalization gap  (stability error), which allows for isolating the effects of optimization dynamics from generalization properties, providing a clearer diagnostic tool for future research.

4. The experiments cover various activation functions, losses, and even preliminary tests on Transformers.

Weaknesses

1. The derived sufficient conditions for vanishing forgetting require extremely aggressive scaling: $m = \tilde{\Omega}(d^8 K^4)$ and $n = \tilde{\Theta}(d^2 K)$. For high-dimensional data with large $d$ and long task sequences  with large $K$, these requirements are computationally prohibitive.

2. The proposition that standard $L_2 $regularization is equivalent to step-size rescaling in the kernel regime is theoretically sound but potentially discouraging. It suggests that popular methods like EWC or SI, which rely on similar regularization principles, might be fundamentally limited in this regime, yet the paper does not propose alternative architectural or algorithmic fixes to overcome this limitation.

3. Assuming the tasks are orthogonal, the analysis is simplified by eliminating distracting terms in the mean vector. In real-world cognitive logic scenarios, tasks often share underlying features or have conflicting gradients due to semantic overlap rather than orthogonality.

---

> ### Author Rebuttal · Authors · 2026-03-31
>
> We thank the reviewer for the careful reading, for their insightful comments, and for highlighting the contributions of our paper. We address the weaknesses and limitations below and we would be happy to clarify any further points.
>
>
> **W1 & L3** - We agree that the sufficient conditions for width in our main theorem are large. In particular, Theorem 1 gives a sufficient scaling for width in the kernel regime, not a claim about the minimal width required in practice. Our view is that these rates mainly reflect the cost of obtaining a fully explicit proof in a nonlinear continual learning setting while keeping the dynamics in the lazy regime. The main value of the theorem is qualitative since it shows how forgetting depends on width, sample size, number of tasks, and training horizon, and shows that vanishing forgetting is achievable in a neural network under explicit conditions. We will emphasize in the revision that our numerical experiments show the same trends at substantially smaller widths than those required by the theorem. Thank you for mentioning this point.
>
> **W2** -  Our claim is more limited than saying that regularization is ineffective in continual learning in general. What we show is that, in the *kernel regime* studied in the paper, the specific *L2-type regularization* around the previous-task solution is equivalent to unregularized continual learning with an effectively smaller step size. Therefore, within this regime, such regularization does not fundamentally improve the scaling laws derived in our results. This still leaves open the possibility that architectural modifications, or regimes with stronger feature learning can improve continual learning in practice. We will revise the manuscript to better reflect this point.
>
> **W3 & L2** - We agree with the reviewer that extensions to non-orthogonal means are interesting. We expect it to be possible within our framework. In fact, we expect the approach is applicable to other data distributions as well. We chose this assumption to obtain a clean characterization of how the main problem parameters enter the forgetting bounds without obscuring the analysis by cross-task interference terms. As noted in the paper, our framework can be extended beyond orthogonal means, but the resulting expressions become more cumbersome because additional correlation terms appear in the dynamic. In fact delayed-generalization gap in our analysis holds for almost any data distribution and our approach for the train-time forgetting shows that for general data, the train-time forgetting is characterized by (cf. the discussion after Thm. 2.2. and  also Eqs. (16)-(17) in the appendix):
> $$|\widehat{F}\_k (w_K)- \widehat{F}\_k (w_k)|=|\sum_{x_k} \eta T x_k^{\top} (\sum_{j=k+1}^K  A_j ) x_k | + O(\frac{|| w_{K T}-w_0 ||^2}{\sqrt{m}}),$$
>
> where we denote $A_j:=\frac{1}{n^2} \sum_{v=1}^n y_v^j x_v^j x_v^{j^{\top}}$. Therefore the bound is applicable to different data distributions. However, we expect the qualitative role of different problem parameters on mitigating forgetting to remain the same. We will include this as a remark in the next version of the paper. Thank you for highlighting this point.
>
> **L1** - Extensions to feature learning regime is highly desirable. Even in the restricted setting we currently have, the paper establishes several nontrivial facts: 1. test-time forgetting can be decomposed into train-time forgetting and delayed generalization gap; 2. vanishing forgetting is possible in a nonlinear neural network without explicit regularization; and 3. later-task sample size can reduce forgetting of earlier tasks. We see an extension to feature-learning regimes as an important and complementary next direction, and we will highlight this more explicitly in the revised paper.

---

> > ### Author Rebuttal · Reviewer_BBCJ · 2026-04-01
> >
> > Thank you for the clarifications.   My concerns have been adequately addressed.

---

> > > ### Author Response · Authors · 2026-04-07
> > >
> > > Thank you for reviewing our rebuttal and for confirming that your concerns have been addressed. We appreciate your insightful feedback, which helped us improve the clarity of the paper.

---

### Official Review · Reviewer_sxMp · 2026-03-14

**Soundness:** 3
**Presentation:** 4
**Significance:** 3
**Originality:** 4
**Overall Recommendation:** 5
**Confidence:** 3

**Summary:**

The paper studies continual learning and catastrophic forgetting from a theoretical perspective. They study the effect of continual learning on a model with one hidden layer on XOR cluster datasets. The paper decomposes 'test-time forgetting' to 'train-time forgetting' and 'delayed generalization gap', and provides upper-bounds for them for the studied model. Based on the results, the authors suggest a scaling in which the continual learning is possible, and they study the role of sample size and width on forgetting. The paper includes several experiments, validating the finding of the theory.

**Compliance With Llm Reviewing Policy:**

Affirmed.

**Final Justification:**

I believe my current score is appropriate.

**Key Questions For Authors:**

- Can you use your model to identify a regime in which without regularization, continual learning is not possible, but with regularization it is?

- In Figure 3, why does the performance on task 1 improves when transitioning from task 2 to 3?

- In the algorithm, it is mentioned at each step a batch of data is used to compute the update. Can you provide the batch size for the experiments? It is missing from the hyperparameters. Specifically, when increasing the number of samples in the experiments is the batch size fixed?

**Limitations:**

Yes

**Strengths And Weaknesses:**

Strengths:

- The problem is very important and interesting for the community and there are not many theoretical works on continual learning.

- The paper gives closed-form forgetting guarantees for a neural network, and given these guarantees a scaling for parameters is suggested that makes continual learning possible.

- Based on the given upper-bound the role of sample size, width and number of training steps is studied.

- The experiments confirm the insights from the theory.

Weaknesses:

- The data model with XOR clusters with orthogonal means is a quite strong data assumption, and the required scaling is also quite strong.

- The setting on the paper does not result to accurate study on the role of regularization consistent with the literature, as there are many works that show the improvement performance in continual learning based on these models.

- There are some gaps in the proof of Theorem B.1 in the Appendix.

- The paper studies gradient descent on tasks sequentially, and con not lead to algorithm design in practice.

---

> ### Author Rebuttal · Authors · 2026-03-31
>
> We thank the reviewer for the careful reading, for their insightful comments, and for highlighting the strengths of our paper. We address the weaknesses and questions below and we would be happy to clarify any further points.
>
> **W1** --  Thank you for your comment on the data model considered in the paper.  We chose this data model because they are a standard non-linearly-separable multi-index model in the deep learning theory literature (cf. the discussion in section 2.1.2  in the paper). That said, an important part of our framework is not restricted to this specific distribution: the delayed generalization-gap analysis is stated for essentially arbitrary task distributions, and several steps in the train-time analysis extend beyond the orthogonal XOR case (see lines after Theorem 2.2. in the paper and the paragraph (starting with W3&L2) in our response to *reviewer BBCJ*). We will clarify this point and emphasize  that orthogonality is mainly used to better characterize  the qualitative roles of different problem parameters in a clean closed form.
>
> **W2** -- Our claim does not really  say  that regularization is ineffective in continual learning in general. What we show is that, in the *kernel regime* studied in the paper, the specific *L2-type regularization* around the previous-task solution is equivalent to unregularized continual learning with an effectively smaller step size. Therefore, within this regime, such regularization does not fundamentally improve the scaling laws derived in our results. This does not preclude other mechanisms (e.g., replay, architectural changes, or regimes with stronger feature learning) from being beneficial in practice. We will revise the discussion to make this point clearer.
>
> **W3** -- We believe the reviewer is referring to the use of Lemma B.5, where we need the leave-one-out training loss to be of the same order as the full-sample training loss. We agree that this assumption should have been stated explicitly in Theorem B.1. Our intention was that this is a mild assumption in the regime studied in the appendix and furthermore it does not affect the proof of our main result in Thm. 2.3.  We will revise the theorem statement to explicitly include this assumption and better clarify the role of Lemma B.5 in the proof. Thank you for pointing this out. Please let us know if any other parts of the proof require clarification.
>
> **W4** -- We agree that the paper is mainly a theoretical paper, not a proposal of a new continual learning algorithm. Our contribution is to provide one of the first explicit forgetting guarantees for neural networks trained sequentially by gradient descent, and to identify how forgetting depends on width, sample size, number of tasks, and training horizon.  We also derived several nontrivial conceptual conclusions: (1) continual learning can be possible without explicit regularization, (2) forgetting admits a clean decomposition into optimization and delayed-generalization components, and (3) later-task sample size can improve retention of earlier tasks.
>
> **Q1** -- Within the kernel regime studied here, our analysis suggests the answer is essentially no for this class of $L_2$ regularization. A regime where regularization can change learnability would likely require leaving the lazy regime, or considering other mechanisms such as replay, architectural constraints. Previous works have also shown empirically  that  different variations of the regularization can lead to improvements for some datasets [e.g. see Kirkpatrick et al’ 17]. Thank you for pointing this out. We will make this point clear in the next version of the paper.
>
> **Q2** -- This behavior can occur because the plotted curves correspond to a single realization of a finite-width, finite-sample experiment, whereas our theory provides upper bounds. Since the later tasks are orthogonal, there is no systematic positive-transfer term that must appear in expectation; however optimization and sampling fluctuations can produce mild non-monotone behavior, including a temporary recovery in task-1 performance. Thank you for this comment.
>
> **Q3** -- Algorithm 1 is written in a general form that includes mini-batch or full-batch updates, but for the theoretical analysis and the experiments reported in the paper we consider full-batch GD. Accordingly, when the number of samples is varied, the effective batch size in those experiments is the full dataset size for that task. We will add this to the experimental details.

---

> > ### Author Rebuttal · Reviewer_sxMp · 2026-04-04
> >
> > Thank you for the clarifications, and I believe my current score is appropriate.

---

> > > ### Author Response · Authors · 2026-04-07
> > >
> > > Thank you for the acknowledgment. We appreciate your insightful comments, which helped improve the clarity of the paper.

---

### Decision · Program_Chairs · 2026-04-30

**Decision:**

Accept (regular)

**Comment:**

While ratings on the paper are split among the reviewers (scores: 5, 5, 4, 2), reviewer jE2c did not engage in the subsequent discussion phase to contest the authors' rebuttal.

The three positive reviewers highlighted the novelty and the difficulty of providing explicit theoretical results for continual learning, even within simplified settings. In particular, the work provides closed-form forgetting guarantees for two-layer quadratic neural networks trained with gradient descent, showing how forgetting decomposes into a training error and a delayed generalisation gap. It also obtains scaling laws for network width, sample size, and the training horizon necessary to achieve vanishing forgetting without regularisation.

Reviewer jE2c (score 2) was against acceptance because of significance, arguing that the results largely align with existing empirical intuitions and that the setting is highly restrictive. However, the authors provided a rebuttal clarifying the assumptions and contextualising the bounds. Given that the remaining concern is highly subjective (significance of proving intuitive results) and the majority of the reviewers actively disagree with this assessment, I side with the majority.

The authors have adequately addressed the technical concerns, and the paper represents a solid theoretical contribution to the continual learning community. I recommend acceptance as a poster.